# Geometric All-Way Boolean Tensor Decomposition

**Changlin Wan**[1,2], **Wennan Chang**[1,2], **Tong Zhao**[3], **Sha Cao**[2], **Chi Zhang**[2]

[1] Purdue University, [2] Indiana University, [3] Amazon

{wan82,chang534}@purdue.edu, zhaoton@amazon.com, {shacao,czhang87}@iu.edu

## Abstract

Boolean tensor has been broadly utilized in representing high dimensional logical data collected on spatial, temporal and/or other relational domains. Boolean Tensor Decomposition (BTD) factorizes a binary tensor into the Boolean sum of multiple rank-1 tensors, which is an NP-hard problem. Existing BTD methods have been limited by their high computational cost, in applications to large scale or higher order tensors. In this work, we presented a computationally efficient BTD algorithm, namely *Geometric Expansion for all-order Tensor Factorization* (GETF), that sequentially identifies the rank-1 basis components for a tensor from a geometric perspective. We conducted rigorous theoretical analysis on the validity as well as algorithemic efficiency of GETF in decomposing all-order tensor. Experiments on both synthetic and real-world data demonstrated that GETF has significantly improved performance in reconstruction accuracy, extraction of latent structures and it is an order of magnitude faster than other state-of-the-art methods.

## 1 Introduction

A tensor is a multi-dimensional array that can effectively capture the complex multidimensional features. A Boolean tensor is a tensor that assumes binary values endowed with the Boolean algebra. Boolean tensor has been widely adopted in many fields, including knowledge graph, recommendation system, spatial-temporal data etc [1, 2, 3, 4, 5]. Tensor decomposition is a powerful tool in extracting meaningful latent structures in the data, for which the popular CANDECOMP/PARAFAC (CP) decomposition is a generalization of the matrix singular value decomposition to tensor [6]. However, these algorithms are not directly usable for Boolean tensors. In this study, we focus on Boolean tensor decomposition (BTD) under similar framework to the CP decomposition.

As illustrated in Figure 1, BTD factorizes a binary tensor $\mathcal{X}$ as the Boolean sum of multiple rank 1 tensors. In cases when the error distribution of the tensor data is hard to model, BTD applied to binarized data can retrieve more desirable patterns with better interpretation than regular tensor decomposition [7, 8]. This is probably due to the robustness of logic representation of BTD. BTD is an NP-hard problem [7]. Existing BTD methods suffer from low efficiency due to high space/time complexity, and particularly, most BTD algorithms adopted a least square updating approach with substantially high computational cost [9, 10]. This has hindered their application to either large scale datasets, such as social network or genomics data, or tensors of high-order.

We proposed an efficient BTD algorithm motivated by the geometric underpinning of rank-1 tensor bases, namely GETF (Geometric Expansion for all-order Tensor Factorization). To the best of our knowledge, GETF is the first algorithm that can efficiently deal with all-order Boolean tensor decomposition with an $O(n)$ complexity, where $n$ represents the total number of entries in a tensor. Supported by rigorous theoretical analysis, GETF solves the BTD problem via sequentially identifying the fibers that most likely coincides with a rank-1 tensor basis component. Our synthetic and real-world data based experiments validated the high accuracy of GETF and its drastically improved efficiency compared with existing methods, in addition to its potential utilization on large scale or high order data, such as complex relational or spatial-temporal data. The key contributions of

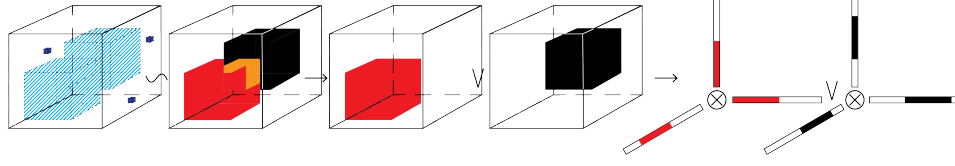

Figure 1: Boolean tensor decomposition

this study include: (1) Our proposed GETF is the first method capable of all-order Boolean tensor decomposition; (2) GETF has substantially increased accuracy in identifying true rank-1 patterns, with less than a tenth of the computational cost compared with state-of-the-art methods; (3) we provided thorough theoretical foundations for the geometric properties for the BTD problem.

## 2 Preliminaries

### 2.1 Notations

Notations in this study follow those in [11]. We denote the *order* of a tensor as $k$, which is also called *ways* or *modes*. Scalar value, vector, matrix, and higher order tensor are represented as lowercase character $x$, bold lowercase character $\mathbf{x}$, uppercase character $X$, and Euler script $\mathcal{X}$, respectively. Super script with mark $\times$ indicates the size and dimension of a vector, matrix or tensor while subscript specifies an entry. Specifically, a $k$-order tensor is denoted as $\mathcal{X}^{m_1 \times m_2 \ldots \times m_k}$ and the entry of position $i_1, i_2, \ldots, i_k$ is represented as $\mathcal{X}_{i_1 i_2 \ldots i_k}$. For a 3-order tensor, we denote its fibers as $\mathcal{X}_{:i_2 i_3}$, $\mathcal{X}_{i_1 : i_3}$ or $\mathcal{X}_{i_1 i_2 :}$ and its slices $\mathcal{X}_{i_1 ::}$, $\mathcal{X}_{:i_2 :}$, $\mathcal{X}_{::i_3}$. For a $k$-order tensor, we denote its mode-$p$ fiber as $\mathcal{X}_{i_1 \ldots i_{p-1} : i_{p+1} \ldots i_k}$ with all indices fixed except for $i_p$. $||\mathcal{X}||$ represents the norm of a tensor, and $|\mathcal{X}|$ the $L_1$ norm in particular. The basic Boolean operations include $\wedge (and, 1 \wedge 1 = 1, 1 \wedge 0 = 0, 0 \wedge 0 = 0)$, $\vee (or, 1 \vee 1 = 1, 1 \vee 0 = 1, 0 \vee 0 = 0)$, and $\neg (not, \neg 1 = 0, \neg 0 = 1)$. Boolean entry-wise sum, subtraction and product of two matrices are denoted as $A \oplus B = A \vee B$, $A \ominus B = (A \wedge \neg B) \vee (\neg A \wedge B)$ and $A \circledast B = A \wedge B$. The outer Boolean product in this paper is considered as the addition of rank-1 tensors, which follows the scope of CP decomposition [6]. Specifically, a three-order Rank-1 tensor can be represented as the Boolean outer product of three vectors, i.e. $\mathcal{X}^{m_1 \times m_2 \times m_3} = \mathbf{a}^{m_1} \otimes \mathbf{b}^{m_2} \otimes \mathbf{c}^{m_3}$. Similarly, for higher order tensor, $\mathcal{X}^{m_1 \times m_2 \ldots \times m_k}$ of rank $l$ is the outer product of $A^{m_1 \times l, 1}, A^{m_2 \times l, 2}, ..., A^{m_k \times l, k}$, i.e. $\mathcal{X}^{m_1 \times m_2 \ldots \times m_k} = \vee_{j=1}^{l}(A_{:j}^{m_1 \times l, 1} \otimes A_{:j}^{m_2 \times l, 2} ... \otimes A_{:j}^{m_k \times l, k})$ and $\mathcal{X}_{i_1 i_2, \ldots, i_k} = \vee_{j=1}^{l}(A_{i_1 j}^{m_1 \times l, 1} \wedge A_{i_2 j}^{m_2 \times l, 2} ... A_{i_k j}^{m_k \times l, k})$, $j = 1...l$ represents the rank-1 tensor components of a rank $l$ CP decomposition of $\mathcal{X}$. In this paper, we denote $A^{m_i \times l, i}$, $i = 1...k$ as the pattern matrix of the $i$th order of $\mathcal{X}$, its $j$th column $A_{:j}^{m_i \times l, i}$ as the $j$th pattern fiber of the $i$th order, and $A_{:j}^{m_1 \times l, 1} \otimes A_{:j}^{m_2 \times l, 2} ... \otimes A_{:j}^{m_k \times l, k}$ as the $j$-th rank-1 tensor pattern.

### 2.2 Problem statement

As illustrated in Figure 1, for a binary $k$-order tensor $\mathcal{X} \in \{0, 1\}^{m_1 \times m_2 \ldots \times m_k}$ and a convergence criteria parameter $\tau$, the Boolean tensor decomposition problem is to identify low rank binary pattern matrices $A^{m_1 \times l, 1*}, A^{m_2 \times l, 2*}, ... A^{m_k \times l, k*}$, the outer product of which best fit $\mathcal{X}$, where $A^{m_1 \times l, 1*}, A^{m_2 \times l, 2*}, ..., A^{m_k \times l, k*}$ are matrices of $l$ columns. In other words, $(A^{m_1 \times l, 1*}, A^{m_2 \times l, 2*}, ... A^{m_k \times l, k*}) = argmin_{A^1, A^2, ..., A^k}(\gamma(A^{m_1 \times l, 1}, A^{m_2 \times l, 2}, ..., A^{m_k \times l, k}; \mathcal{X})|\tau)$ Here $\gamma(A^{m_1 \times l, 1}, A^{m_2 \times l, 2}, ... A^{m_k \times l, k}; \mathcal{X})$ is the cost function. In general, $\gamma$ is defined to the reconstruction error $\gamma(A^{m_1 \times l, 1*}, ... A^{m_k \times l, k*}; \mathcal{X}) = ||\mathcal{X} \ominus (A^{m_1 \times l, 1*} \otimes ... \otimes A^{m_k \times l, k*})||_{L_p}$, and $p$ is usually set to be 1.

### 2.3 Related work

In order of difficulty, Boolean tensor decomposition consists of three major tasks, Boolean matrix factorization (BMF, $k = 2$) [12], three-way Boolean tensor decomposition (BTD, $k = 3$) and higher order Boolean tensor decomposition (HBTD, $k > 3$) [13]. All of them are NP hard [7]. Numerous heuristic solutions for the BMF and BTD problems have been developed in the past two decades [9, 14, 15, 16, 17, 18, 19, 20].

For BMF, the ASSO algorithm is the first heuristic BMF approach that finds binary patterns embedded within row-wise correlation matrix [14]. On another account, PANDA [19] sequentially retrieves patterns under current (residual) binary matrix amid noise. Recently, BMF algorithms using Bayesian framework have been proposed [8]. The latent probability distribution adopted by Message Passing (MP) achieved top performance among all the state-of-the-art methods for BMF [21].

For BTD, Miettinen et al thoroughly defined the BTD problem ($k = 3$) in 2011 [15], and proposed the use of least square update as a heuristic solution. To solve the scalability issue, they further developed Walk'N'Merge, which applies random walk over a graph in identifying dense blocks as proxies of rank 1 tensors [17]. Despite the increase of scalability, Walk'N'Merge tends to pick up many small patterns, the addition of which doesn't necessarily decrease the loss function by much. The DBTF algorithm introduced by Park et al. is a parallel distributed implementation of alternative least square update based on Khatri-Rao matrix product [10]. Though DBTF reduced the high computational cost, its space complexity increases exponentially with the increase of tensor orders due to the khatri-Rao product operation. Recently, Tammo et al. proposed a probabilistic solution to BTD, called Logistical Or Machine (LOM), with improved fitting accuracy, robustness to noises, and acceptable computational complexity [2]. However, the high number of iterations it takes to achieve convergence of the likelihood function makes LOM prohibitive to large data analysis. Most importantly, to the best of our knowledge, none of the existing algorithms are designed to handle the HBTD problem for higher order tensors.

## 3 GETF Algorithm and Analysis

GETF[1] identifies the rank-1 patterns sequentially: it first extracts one pattern from the tensor; and the subsequent patterns will be extracted sequentially from the residual tensor after removing the preceding patterns. We first derive the theoretical foundation of GETF. We show that the geometric property of the largest rank-1 pattern in a binary matrix developed in [1] can be naturally extended to higher order tensor. We demonstrated the true pattern fiber of the largest pattern can be effectively distinguished from fibers of overlapped patterns or errors by reordering the tensor to maximize its overlap with a left-triangular-like tensor. Based on this idea, the most likely pattern fibers can be directly identified by a newly develop geometric folding approach that circumvents heuristic greedy searching or alternative least square based optimization.

### 3.1 Theoretical analysis

We first give necessary definitions of the slice, re-order and sum operations on a $k$ order tensor and an theoretical analysis of the property of a left-triangular-like (LTL) tensor.

**Definition 1.** *(p-order slice). The p-order slice of a tensor $\mathcal{X}^{m_1 \times \cdots \times m_k}$ indexed by $\mathbb{P}$ is defined by $\mathcal{X}_{i_1,\ldots,i_k}$, where $i_k$ is a fixed value $\in \{1, \ldots, m_k\}$ if $k \in \bar{\mathbb{P}}$, and $i_k$ is unfixed ($i_k =:$) if $k \in \mathbb{P}$, here $p = |\mathbb{P}|$ and $\bar{\mathbb{P}} = \{1, \ldots, k\} \setminus \mathbb{P}$. Specifically, we denote a $|\mathbb{P}|$ order slice of $\mathcal{X}^{m_1 \times \cdots \times m_k}$ with the index set $|\mathbb{P}|$ unfixed as $\mathcal{X}_{\mathbb{P}, I_{\bar{\mathbb{P}}}}^{m_1 \times \cdots \times m_k}$ or $\mathcal{X}_{\mathbb{P}, I_{\bar{\mathbb{P}}}}$, in which $\mathbb{P}$ is the unfixed index set and $I_{\bar{\mathbb{P}}}$ are fixed indices.*

**Definition 2.** *(Index Reordering Transformation, IRT). The index reordering transformation (IRT) transforms a tensor $\mathcal{X}^{m_1 \times, \cdots \times m_k}$ to $\tilde{\mathcal{X}} = \mathcal{X}_{P_1, P_2, \ldots, P_k}$, where $P_1, \ldots, P_k$ are any permutation of the index sets, $\{1, \ldots, m_1\}, \ldots, \{1, \ldots, m_k\}$.*

**Definition 3.** *(Tensor slice sum). The tensor slice sum of a k-order tensor $\mathcal{X}^{m_1 \times \cdots \times m_k}$ with respect to the index set $\mathbb{P}$ is defined as $T_{sum}(\mathcal{X}, \mathbb{P}) \triangleq \sum_{i_1=1}^{m_{i_1}} \cdots \sum_{i_{|\mathbb{P}|}=1}^{m_{i_{|\mathbb{P}|}}} \mathcal{X}_{:\ldots:i_1:\ldots:i_{|\mathbb{P}|}:\ldots:}, i_1, \ldots, i_{|\mathbb{P}|} \in \mathbb{P}$. $T_{sum}(\mathcal{X}, \mathbb{P})$ results in a $k - |\mathbb{P}|$ order tensor.*

**Definition 4.** *(p-left-triangular-like, p-LTL). A k-order tensor $\mathcal{X}^{m_1 \times \cdots \times m_k}$ is called p-LTL, if any of its p-order slice, $\mathcal{X}_{\mathbb{P}, I_{\bar{\mathbb{P}}}}$, $\mathbb{P} \subset \{1, \ldots, k\}$ and $|\mathbb{P}| = p$, and $\forall j \in \mathbb{P}, 1 \leq j_1 < j_2 \leq m_j$, $T_{sum}(\mathcal{X}_{\mathbb{P}, I_{\bar{\mathbb{P}}}}, \mathbb{P} \setminus \{j\})_{j_1} \leq T_{sum}(\mathcal{X}_{\mathbb{P}, I_{\bar{\mathbb{P}}}}, \mathbb{P} \setminus \{j\})_{j_2}$.*

**Definition 5.** *(flat 2-LTL), A k-order 2-LTL tensor $\mathcal{X}^{m_1 \times \cdots \times m_k}$ is called flat 2-LTL within an error range $\epsilon$, if any of its 2-order slice, $\mathcal{X}_{\mathbb{P}, I_{\bar{\mathbb{P}}}}$, $\mathbb{P} \subset \{1, \ldots, k\}$ and $|\mathbb{P}| = p$, and $\forall j \in \mathbb{P}, 1 \leq j_1 < j_2 \leq m_j$, $|T_{sum}(\mathcal{X}_{\mathbb{P}, I_{\bar{\mathbb{P}}}}, \mathbb{P} \setminus \{j\})_{j_1} + T_{sum}(\mathcal{X}_{\mathbb{P}, I_{\bar{\mathbb{P}}}}, \mathbb{P} \setminus \{j\})_{j_2} - 2T_{sum}(\mathcal{X}_{\mathbb{P}, I_{\bar{\mathbb{P}}}}, \mathbb{P} \setminus \{j\})_{(j_1+j_2)/2}| < \epsilon$.*

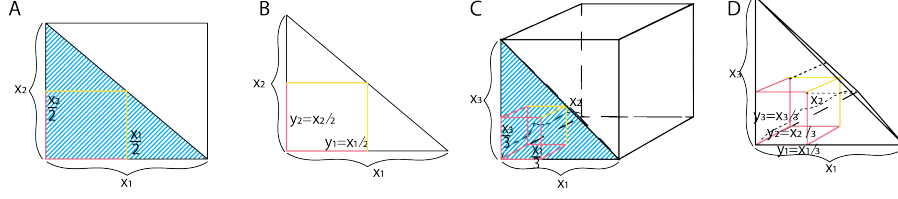

Figure 2: Imitating flat 2-LTL tensor with corresponding geometric object revealed that the basis of optimal rank 1 subarray is seeded on the $1/k$ segmentation point.

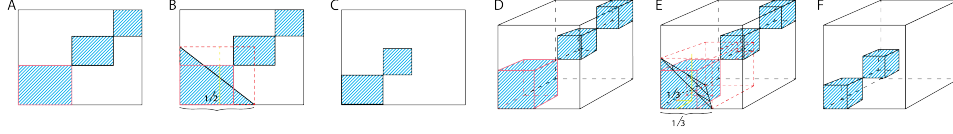

Figure 3: GETF sequentially decompose $k - 1$ LTL tensor with geometric consideration.

The **Definition 5** indicates the tensor sum of over any 2-order slice of a flat 2-LTL tensor is close enough to a linear function with the largest error less than $\epsilon$. Figure 2A,C illustrate two examples of flat 2-$LTL$ matrix and 2-$LTL$ 3-order tensor. By the definition, the non-right angle side of a flat 2-$LTL$ $k$-order tensor is close to a $k - 1$ dimension plane, which is specifically called as the **k-1 dimension plane** of the flat 2-$LTL$ tensor in the rest part of this paper.

**Lemma 1** (Geometric segmenting of a flat 2-$LTL$ tensor)**.** *Assume $\mathcal{X}$ is a $k$-order flat 2-LTL tensor and $\mathcal{X}$ has none zero fibers. Then the largest rank-1 subarray in $\mathcal{X}$ is seeded where one of the pattern fibers is paralleled with the fiber that anchored on the $1/k$ segmenting point (entry $\{|m_1|/k, |m_2|/k, ..., |m_k|/k\}$) along the sides of the right angle.*

**Lemma 2.** *(Geometric perspective in seeding the largest rank-1 pattern) For a $k$ order tensor $\mathcal{X}$ sparse enough and a given tensor size threshold $\lambda$, if its largest rank-1 pattern tensor is larger than $\lambda$, the IRT that reorders $\mathcal{X}$ into a (k-1)-LTL tensor reorders the largest rank-1 pattern to a consecutive block, which maximize the size of the connected solid shape overlapped with the $k - 1$ dimension plane over a flat 2-LTL tensor larger than $\lambda$.*

**Lemma 3.** *If a $k$-order tensor $\mathcal{X}^{m_1 \times ... \times m_k}$ can be transformed into a p-LTL tensor by IRT, the p-LTL tensor is unique, p=2,...,k-1.*

**Lemma 4.** *If a $k$-order tensor is p-LTL, then it is x-LTL, for all the x=p,p+1,...,k.*

Detailed proofs of the **Lemma 1-4** are given in APPENDIX section 2. **Lemma 1** and **2** reflect our geometric perspective in identifying the largest rank-1 pattern and seeding the most likely pattern fibers. Specifically, **Lemma 1** suggests the optimal position of the fiber that is most likely the pattern fiber of the largest rank-1 pattern tensor under a flat 2-$LTL$ tensor. Figure 2B,D illustrate the position (yellow dash lines) of the most likely pattern fibers in the flat 2-$LTL$ matrix and 3-order tensor. It is noteworthy that the (k-1)-$LTL$ tensor must exists for a $k$-order tensor, which can be simply derived by reordering the indices of each tensor order $j$ by the decreasing order of $T_{sum}(\mathcal{X}, \{1, ..., k\} \setminus \{j\})$. However, not all $k$ order tensor can be transformed into a 2-$LTL$ tensor via IRT when $k > 2$. A (k-1)-$LTL$ tensor with only one rank-1 pattern tensor is 2-$LTL$. Intuitively, the left bottom corner of a $k$-order (k-1)-$LTL$ tensor of the largest rank-1 pattern is also 2-$LTL$ (Figure 2D). However, the existence of multiple rank-1 patterns, overlaps among patterns and errors limit the 2-$LTL$ property of left bottom corner of its (k-1)-$LTL$ tensor. **Lemma 2** suggests the indices of the largest rank-1 pattern form the largest overlap between the (k-1)-$LTL$ IRT and the the $k - 1$ dimension plane over a flat 2-$LTL$ tensor. Based on this property, the largest rank-1 pattern and its most likely fiber can be seeded without heuristic greedy search or likelihood optimization that can substantially improve the computational efficiency. **Lemma 3** and **4** suggest that the (k-1)-$LTL$ tensor is the IRT of $\mathcal{X}$ that is closest to a 2-$LTL$ tensor. Hence how close the intersect between a (k-1)-$LTL$ tensor and a 2-$LTL$ sub tensor is to a 2-$LTL$ tensor, can reflect if the optimal pattern fiber position derived in **Lemma 1** fits to the 2-$LTL$ sub tensor region of the (k-1)-$LTL$ tensor.

## 3.2 GETF algorithm

Based on the geometric property of the largest rank-1 pattern and its most likely pattern fibers, we developed an efficient BTD and HBTD algorithm—GETF, by iteratively reconstructing the to-be-decomposed tensor into a $k-1$ LTL tensor and identifying the largest rank-1 pattern. The main algorithm of GETF is formed by the iteration of the following five steps. *Step 1*: For a given tensor $\mathcal{X}^{m_1 \times m_2 \dots \times m_k}$, in each iteration, GETF first reorders the indices of the current tensor into a (k-1)-$LTL$ tensor by IRT (Figure 3A,D); *Step 2*: GETF utilizes **2_LTL_projection** algorithm to identify the flat 2-$LTL$ tensor that maximizes the overlapped region between its $k-1$ dimension plane and current (k-1)-$LTL$ tensor (Figure 3B,E); *Step 3*: A **Pattern_fiber_finding** algorithm is applied to identify the most likely pattern fiber of the overlap region of the 2-$LTL$ tensor and the (k-1)-$LTL$ tensor, i.e., the largest rank-1 pattern (Figure 4); *Step 4*: A **Geometric_folding** algorithm is applied to reconstruct the rank-1 tensor component from the identified pattern fiber that best fit the current to-be-decomposed tensor (Figure 5); and *Step 5*: Remove the identified rank-1 tensor component from the current to-be-decomposed tensor (Figure 3C,F). The inputs of GETF include the to-be-decomposed tensor $\mathcal{X}$, a noise tolerance threshold $t$ parameter, a convergence criterion $\tau$ and a pattern fiber searching indicator $Exha$.

---

**Algorithm 1:** GETF

---

**Inputs:** $\mathcal{X} \in \{0,1\}^{m_1 \times m_2 \dots \times m_k}$, $t \in (0,1)$, $\tau$, $Exha \in \{0,1\}$
**Outputs:** $A^{1*} \in \{0,1\}^{m_1 \times l}$, $A^{2*} \in \{0,1\}^{m_2 \times l}$, ... $A^{k*} \in \{0,1\}^{m_k \times l}$
$GETF(\mathcal{X}, t, \tau, Exha)$:
$\mathcal{X}^{\text{Residual}} \leftarrow \mathcal{X}$, $A^1 \leftarrow NULL$,..., $A^k \leftarrow NULL$
$\Omega \leftarrow$ Generate set of directions for geometric-folding$(k, Exha)$
**while** $!\tau$ **do**
    $\gamma_0 \leftarrow inf$, $\mathbf{a}^{1*} \leftarrow NULL$,..., $\mathbf{a}^{k*} \leftarrow NULL$
    **for** *each direction **o** in* $\Omega$ **do**
        $\mathcal{X}^{2-LTL} \leftarrow$ **2_LTL_projection**$(\mathcal{X})$
        $Pattern\ fiber^* \leftarrow$ **Pattern_fiber_finding**$(\mathcal{X}^{2-LTL}, \mathbf{o})$
        $(\mathbf{a}^1, ..., \mathbf{a}^k) \leftarrow$ **Geometric_folding**$(\mathcal{X}^{Residual}, Patternfiber^*, \mathbf{o}, t)$
        **if** $\gamma(\mathbf{a}^1, ..., \mathbf{a}^k | \mathcal{X}) < \gamma_0$ **then**
            $(\mathbf{a}^{1*}, ..., \mathbf{a}^{k*}) \leftarrow (\mathbf{a}^1, ..., \mathbf{a}^k)$; $\gamma_0 \leftarrow \gamma(\mathbf{a}^1, ..., \mathbf{a}^k | \mathcal{X})$
    **end**
    **if** $\gamma_0 \neq inf$ **then**
        $\mathcal{X}^{Residual}_{i_1 i_2 \dots i_k} \leftarrow 0$ when $(\mathbf{a}^{1*} \otimes \mathbf{a}^{2*} \dots \otimes \mathbf{a}^{k*})_{i_1 i_2 \dots i_k} = 1$
        $A^{j*} \leftarrow append(A^{j*}, \mathbf{a}^{j*})$, $j \in \{1, 2, ..., k\}$
**end**

---

Details of the GETF and its sub algorithms are given in APPENDIX section 3. In **Algorithm 1**, **o** represents a direction of geometric folding, which is a permutation of $\{1, ..., k\}$. The **2_LTL_projection** utilizes a project function and a scoring function to identify the flat 2-$LTL$ tensor that maximizes the solid overlapped region between its $k-1$ dimension plane and a (k-1)-$LTL$ tensor. The **Pattern_fiber_finding** and **Geometric_folding** algorithm are described below. Noted, there are $k$ directions of pattern fibers and $k!$ combinations of the orders in identifying them from a $k$-order tensor or reconstructing a rank-1 pattern from them. Empirically, to avoid duplicated computations, we tested conducting $k$ times of geometric folding is sufficient to identify the fibers and reconstruct the suboptimal rank-1 pattern. GETF also provides options to set the rounds and noise tolerance level of geometric folding in expanding a pattern fiber via adjusting the parameters $Exha$ and $t$.

## 3.3 Pattern fiber finding

The **Pattern_fiber_finding** algorithm is developed based on **Lemma 1**. Its input include a $k$-order tensor and a direction vector. Even the input is the entry-wise product of a flat 2-$LTL$ tensor and the largest rank-1 pattern in a (k-1)-$LTL$ tensor, it may still not be 2-$LTL$ due to the existence of errors. We propose a recursive algorithm that recurrently rearrange an order of the input tensor and reports the coordinate of the pattern fiber on this order (See details in APPENDIX section 3.5). The output is the position of the pattern fiber.

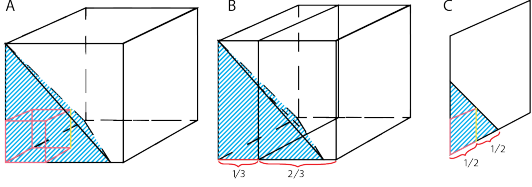

Figure 4: Illustration of finding pattern fiber in 3D

Here in Figure 4, we illustrates the pattern fiber finding approach for a 3-order flat 2-$LTL$ tensor $\mathcal{X}^{m_1 \times m_2 \times m_3}$. To identify the coordinates of the yellow colored pattern fiber with unfixed index of the 1st order $\mathcal{X}_{:i_2 i_3}$ (Figure 4A), its coordinate of the 2nd order is anchored on the 1/3 segmenting point of $T_{sum}(\mathcal{X}, \{2\})$, denoted as $i_2$ (Figure 4B), and its coordinate of the 3rd order is on the 1/2 segmenting point of $T_{sum}(\mathcal{X}_{:i_2:}^{m_1 \times m_2 \times m_3}, \{2\})$ (Figure 4C).

## 3.4 Geometric folding

The geometric folding approach is to reconstruct the rank-1 tensor pattern best fit $\mathcal{X}$ from the pattern fiber identified by the **Pattern_fiber_finding** algorithm (see details in APPENDIX section 3.6). For a $k$-order tensor $\mathcal{X}$ and the identified position of pattern fiber, the pattern fiber is

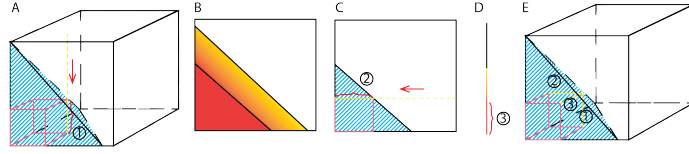

Figure 5: Illustration of Geometric_folding in 3D

denote as $\mathcal{X}_{:i_2^0...i_k^0}$ (Figure 5A). The algorithm computes the inner product between $\mathcal{X}_{:i_2^0...i_k^0}$ and each fiber $\mathcal{X}_{:i_2...i_k}$ to generate a new $k-1$ order tensor $\mathcal{H}^{m_2 \times ... \times m_k}$, $\mathcal{H}_{i_2...i_m} = \sum_{j=1}^{m_1} \mathcal{X}_{ji_2^0...i_k^0} \wedge \mathcal{X}_{ji_2...i_k}$ (Figure 5B). This new tensor is further discretized based on a user defined noise tolerance level and generates a new binary $k$-1 order tensor $\mathcal{X}'^{m_2 \times m_3... \times m_k}$ (Figure 5C). This approach is called as geometric folding of a $k$-order tensor into a $k$-1 order tensor based on the pattern fiber $\mathcal{X}_{:i_2^0...i_k^0}$. This approach will be iteratively conducted to fold the $k$-way tensor into a 2 dimensional matrix with $k$-2 rounds of **Pattern_fiber_finding** and **Geometric_folding** and identifies $k$-2 pattern fibers. The pattern fibers of the last 2 dimensional will be identified as a BMF problem by using MEBF [1]. The output of **Geometric_folding** is the set of $k$ pattern fibers of a rank-1 tensor (Figure 5E).

## 3.5 Complexity analysis

Assume $k$-order tensor has $n = m^k$ entries. The computation of **2_LTL_projection** is fixed based on its screening range, which is smaller than $O(m^k)$. To further accelerate GETF, we omitted the projection step as it does not affect the overall decomposition efficiency in practise. The computation of each **Pattern_fiber_finding** is $\frac{m^{k+1}-m}{m-1} + km\log(m)$. **Geometric_folding** is a loop algorithm consisted of additions and **Pattern_fiber_finding**. The computation for **Geometric_folding** to fold a $k$-order tensor takes $\frac{2m^{k+2}-m^k}{(m-1)^2} + \frac{1-2m^2}{(m-1)^2} - \frac{km}{m-1} + \frac{k(k+1)}{2}m\log(m)$ computations. $GETF$ conducts $k$ times $Geometric\_folding$ in each iteration to extract the suboptimal rank-1 tensor, by which, the overall computing cost on each iteration is $k(\frac{2m^{k+2}-m^k}{(m-1)^2} + \frac{1-2m^2}{(m-1)^2} - \frac{km}{m-1} + \frac{k(k+1)}{2}m\log(m)) \sim O(m^k)$. Hence $GETF$ is an $O(m^k) = O(n)$ complexity algorithm. More detailed complexity analysis can be seen in APPENDIX section 3.

## 4 Experimental Results on Synthetic Datasets

We generated synthetic tensors with $k = 2, 3, 4, 5$ that correspond to the BMF, BTD, 4-order HBTD and 5-order HBTD problems, and for each order $k$, 4 scenarios are further created: (1) low density tensor without error, (2) low density tensor with error, (3) high density tensor without error and (4) high density tensor with error. Under each scenario, we fixed the number of true patterns as 5 and set the convergence criteria as 1) 10 patterns have been identified, 2) the cost function stopped decreasing with newly identified patterns. Detailed experiment setup is listed in APPENDIX section 4. We compared GETF with MP on BMF and LOM on BTD settings, which represent best performance for BMF and BTD problems respectively [2, 21]. The evaluation focuses on two metrics, time

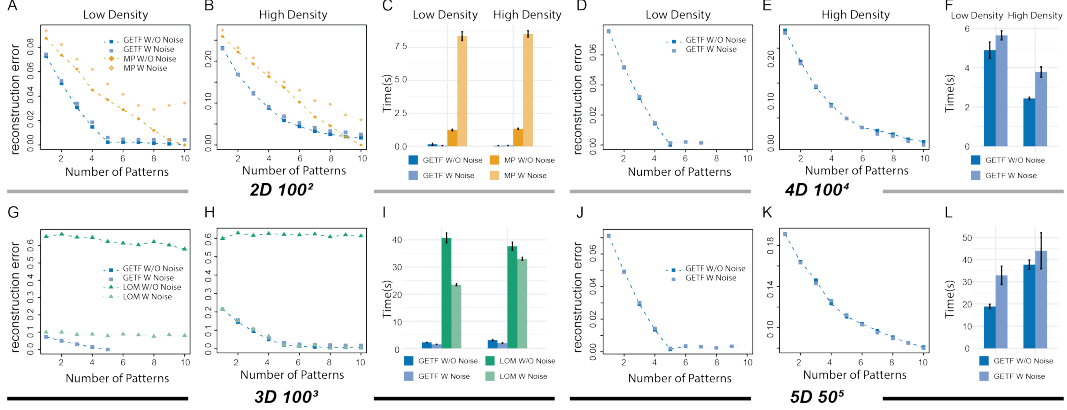

Figure 6: Performance analysis on simulated data

consumption and reconstruction error [8, 21]. For 4-order and 5-order HBTD, we only conducted GETF as other methods cannot handle or fail to converge in analyzing such high order tensors.

GETF significantly outperformed MP in reconstruction error (Figure 6A,B) and time consumption (Figure 6C) for all the four scenarios. Noted, the top five patterns identified by GETF coincided with the five true patterns in all scenarios. Similarly GETF outperformed LOM in all four scenarios, except for the high density with high noise case, where GETF and LOM performed comparatively in terms of reconstruction error (Figure 6G,H,I). We also evaluated each algorithm on different data scales, as detailed in APPENDIX section 4. GETF maintains the most favorable performance with over 10 times higher in computational efficiency. Figure 6 D-F,J-L show the capability of GETF on decomposing high order tensor data. Notably, the reconstruction error curve of GETF flattened after reaching the true number of components (Figure 6A,B,D,E,G,H,J,K), suggesting its high accuracy in identifying true number of patterns. The error bar stands for standard derivation of time consumption in Figure 6 C,F,I,L. Importantly, when the tensor order increases, its size increases exponentially. The high memory cost is regarded as an outstanding challenge for higher order tensor decomposition, for which an $O(n)$ algorithm like GETF is desirable. GETF showed consistent performance with respect to different noise levels. For a 5-way tensor with more than $3 * 10^8$ elements, GETF reached convergence in less than 1 minute. Overall, our experiments on synthetic datasets advocated the efficiency and robustness of GETF for the data with different tensor orders, data sizes, signal densities and noise levels.

## 5 Experimental Results on Real-world Datasets

We applied GETF on two real-world datasets, the Chicago crime record data[2], and a breast cancer spatial-transcriptomics data[3], which represents two scenarios with relatively lower and higher noise. We benchmarked GETF with LOM on the two data sets, and focused on comparing their reconstruction error and interpreting the pattern tensors identified by GETF. Detailed benchmark were provided in APPENDIX section 5.

We retrieved the crime records in Chicago from 2001 to 2019 and organized them into a 4D tensor, with four dimensions representing: 436 regions, 365 dates, 19 years and two crime categories (severe, and non-severe), respectively, i.e., $\mathcal{X}^{436 \times 365 \times 19 \times 2}$. An entry in the tensor has value 1 if a crime category was observed in the region for the day of the year. We first benchmark GETF with LOM on the 3D slice, $\mathcal{X}_{:::1} \in \{0, 1\}^{436 \times 365 \times 19}$. GETF showed a clear advantage over LOM, including a less reconstruction error and higher interpretability. GETF converged after identifying two large patterns, while LOM identied more than eight patterns to achieve same level of reconstruciton error (Figure 7B). We only presented the results of GETF on the application of the 4-order tensor, on which LOM failed to converge, and used the top two patterns to reconstruct the original tensor, denoted as $\mathcal{X}^*$. To look for the crime date pattern, the crime index of a region defined as the total days of a year with

crime occurrences in the region were associated with the identified low rank tensor patterns. We showed that $\mathcal{X}^*$ reconstructed from the CP decomposition is a denoised form of the origincal data. In Figure 7C, the high and low crime index were red and blue colored. Clearly, GETF reconstructed tensor is able to distinguish the two regions (Figure 7C). However, such a clear separation is less clear in the original data (Figure 7D). Next we examined the validity of the two regions with an outsider factor, regional crime counts, defined as the total number of crimes from 2001 to 2019 for that region. As shown in Figure 7E, the regions with higher crime index according to GETF consitently correspond to the regions of higher regional crime counts, and vice versa. In summary, we show that GETF is able to reveal the overall crime patterns by denoising the original tensor (see details in APPENDIX section 5.1).

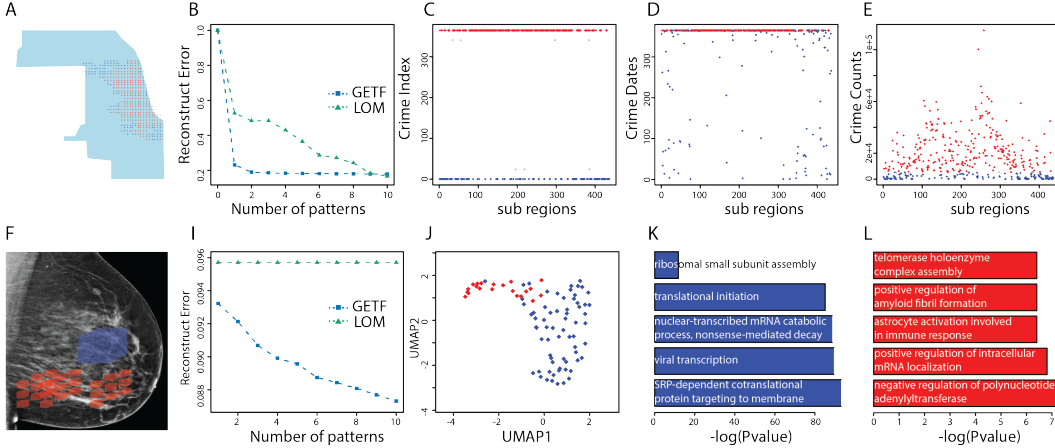

Figure 7: Real data benchmark and applications (see figure information in APPENDIX section 5)

The breast cancer spatial transcriptomics dataset [22, 23], as in Figure 7F, was collected on a 3D space with 1020 cell positions ($x \times y \times z = 15 \times 17 \times 4$), each of which has expression values of 13,360 genes, i.e., $\mathcal{X}^{13360 \times 15 \times 17 \times 4}$. The tensor was first binarized, and it takes value 1 if the expression of the gene is larger than zero. We again benchmarked the performance of GETF and LOM on a 3D slice, $\mathcal{X}_{:::1}$. LOM failed to generate any useful information seen from the non-decreasing reconstruction error, possibly because of the high noise of the transcriptomics data. On the other hand, GETF manage to derive patterns gradually (Figure 7I). We applied GETF only to the 4D tensor, and among the top 10 patterns, we analyzed two extremest patterns: one the most sparse (red) and the other the most dense (blue) (Figure 7F). The sparse pattern has 24 cell positions all expressing 232 genes ($232 \times 4 \times 4 \times 2$), the dense pattern has 90 cells positions expressing 40 genes ($40 \times 15 \times 3 \times 2$). A lower dimensional embedding of the 114 cells using UMAP [24] demonstrated them to be two distinct clusters (Figure 7J). We also conducted functional annotations using gene ontology enrichment analysis for the genes of the two patterns. Figure 7K,L showed the $-log(p)$ of the top 5 pathways enriched by the genes in each pattern, assessed by hypergeometric test. It implies that genes in the most dense pattern maintains the vibrancy of the cancer by showing strong activities in transcription and translation; while genes in the most sparse pattern maintains the tissue structure and suppress anti-tumor immune effect. Our analysis demonstrated that the GETF is able to reveal the complicated but integrated spatial structure of breast cancer tissue with different functionalities.

# 6  Conclusion and Future Work

In this paper, we proposed GETF as the first efficient method for the all-way Boolean tensor decomposition problem. We provided rigorous theoretical analysis on the validity of GETF and conducted experiments on both synthetic and real-world datasets to demonstrate its effectiveness and computational efficiency. In the future, to enable the integration of prior knowledge, we plan to enhance GETF with constrained optimization techniques and we believe it can be beneficial for broader applications that desire a better geometric interpretation of the hidden structures.

## 7 Broader Impact

GETF is a Boolean tensor factorization algorithm, which provides a solution to a fundamental mathematical problem. Hence we consider it is not with a significant subjective negative impact to the society. The structure of binary data naturally encodes the structure of subspace clusters in the data structure. We consider the efficient BTD and HBTD capability led by GETF enables the seeding of patterns for subspace clustering identification or disentangled representation learning, for the data with unknown subspace structure, such as recommendation of different item classes to customers with unknown groups or biomedical data of different patient classes. As we have demonstrated the high computational efficiency of GETF grants the capability to analyze large or high order tensor data, another field can be potentially benefited by GETF is the inference made to the spatial-temporal data collected from mobile sensors. The high efficiency of GETF enable a possible implementation on smart phones for a real-time inference of the data collected from the phones or other multi-modal personal wearable sensors.

## 8 Funding Disclosure

This work was supported by R01 award #1R01GM131399-01, NSF IIS (N0.1850360), Showalter Young Investigator Award from Indiana CTSI and Indiana University Grand Challenge Precision Health Initiative.

## Footnotes

[1]Code can be accessed at https://github.com/clwan/GETF

[2]Chicago crime records downloaded on March 1st, 2020 from https://data.cityofchicago.org/Public-Safety

[3]Breast cancer data is retrieved from https://www.spatialresearch.org/resources-published-datasets

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
