[Supplementary Material]

# Geometric All-Way Boolean Tensor Decomposition APPENDIX

**Changlin Wan**[1,2], **Wennan Chang**[1,2], **Tong Zhao**[3], **Sha Cao**[2], **Chi Zhang**[2]

[1] Purdue University, [2] Indiana University, [3] Amazon

{wan82,chang534}@purdue.edu, zhaoton@amazon.com, {shacao,czhang87}@iu.edu

## 1 Definitions

For ease of illustration, we first recap the definitions that appeared in main text.

**Definition 1.** *(p-order slice). A p-order slice of a $k$ order tensor $\mathcal{X}^{m_1 \times \cdots \times m_k}$ with fixed index set $\mathbb{P}$ ($p = |\mathbb{P}|$) is defined by $\mathcal{X}_{i_1,\ldots,i_k}$, where $i_k$ is a fixed value $\in \{1,\ldots,m_k\}$ if $k \in \mathbb{P}$, and $i_k$ is unfixed ($i_k =:$) if $k \in \bar{\mathbb{P}}$. Specifically, we denote a $|\mathbb{P}|$ order slice of $\mathcal{X}^{m_1 \times \cdots \times m_k}$ with the index set $|\mathbb{P}|$ unfixed as $\mathcal{X}^{m_1 \times \cdots \times m_k}_{\mathbb{P},I_{\bar{\mathbb{P}}}}$ or $\mathcal{X}_{\mathbb{P},I_{\bar{\mathbb{P}}}}$, in which $\mathbb{P}$ is the unfixed index set and $I_{\bar{\mathbb{P}}}$ are fixed indices.*

**Definition 2.** *(Index Reordering Transformation (IRT)). The IRT of a $k$-order tensor $\mathcal{X}^{m_1 \times \cdots \times m_k}$ is defined by $\tilde{\mathcal{X}}_{P_1 P_2, \ldots, P_k}$, where $P_1, \ldots, P_k$ are any permutation of the index sets, $\{1,\ldots,m_1\},\ldots,\{1,\ldots,m_k\}$.*

**Definition 3.** *(tensor slice sum), The tensor slice sum of a $k$-order tensor $\mathcal{X}^{m_1 \times \cdots \times m_k}$ with respect to the index set $\mathbb{P}$ is defined as $T_{sum}(\mathcal{X},\mathbb{P}) \triangleq \sum_{i_1=1}^{m_{i_1}} \cdots \sum_{i_{|\mathbb{P}|}=1}^{m_{i_{|\mathbb{P}|}}} \mathcal{X}_{:\ldots:i_1:\ldots:i_{|\mathbb{P}|}:\ldots:}, i_1,\ldots,i_{|\mathbb{P}|} \in \mathbb{P}$. $T_{sum}(\mathcal{X},\mathbb{P})$ results in a $k - |\mathbb{P}|$ order tensor.*

**Definition 4.** *(p-left triangular like (p-LTL), A $k$-order tensor $\mathcal{X}^{m_1 \times \cdots \times m_k}$ is called p-LTL, if any of its p-order slice, $\mathcal{X}_{\mathbb{P},I_{\bar{\mathbb{P}}}}, \mathbb{P} \subset \{1,\ldots,k\}$ and $|\mathbb{P}| = p$, and $\forall j \in \mathbb{P}, 1 \leq j_1 < j_2 \leq m_j$, $T_{sum}(\mathcal{X}_{\mathbb{P},I_{\bar{\mathbb{P}}}}, \mathbb{P} \setminus \{j\})_{j_1} \leq T_{sum}(\mathcal{X}_{\mathbb{P},I_{\bar{\mathbb{P}}}}, \mathbb{P} \setminus \{j\})_{j_2}$.*

**Definition 5.** *(flat 2-LTL), A $k$-order 2-LTL tensor $\mathcal{X}^{m_1 \times \cdots \times m_k}$ is called flat 2-LTL within an error range $\epsilon$, if any of its 2-order slice, $\mathcal{X}_{\mathbb{P},I_{\bar{\mathbb{P}}}}, \mathbb{P} \subset \{1,\ldots,k\}$ and $|\mathbb{P}| = p$, and $\forall j \in \mathbb{P}, 1 \leq j_1 < j_2 \leq m_j$, $|T_{sum}(\mathcal{X}_{\mathbb{P},I_{\bar{\mathbb{P}}}}, \mathbb{P} \setminus \{j\})_{j_1} + T_{sum}(\mathcal{X}_{\mathbb{P},I_{\bar{\mathbb{P}}}}, \mathbb{P} \setminus \{j\})_{j_2} - 2T_{sum}(\mathcal{X}_{\mathbb{P},I_{\bar{\mathbb{P}}}}, \mathbb{P} \setminus \{j\})_{(j_1+j_2)/2}| < \epsilon$.*

## 2 Lemma

**Lemma 1** (Geometric segmenting of a flat $2$-$LTL$ tensor). *Assume $\mathcal{X}$ is a $k$-order flat $2$-$LTL$ tensor and $\mathcal{X}$ has none zero fibers. Then the largest rank-1 subarray in $\mathcal{X}$ is seeded where one of the pattern fibers is paralleled with the fiber that anchored on the $1/k$ segmenting point (entry $\{|m_1|/k, |m_2|/k, \ldots, |m_k|/k\}$) along the sides of the right angle.*

Figure 1A,C illustrate flat 2-LTL matrix and 3-order tensor. Figure 1B,D illustrate the position (yellow dash lines) of the most likely pattern fibers in the flat $2$-$LTL$ matrix and 3-order tensor. We will first prove Lemma 1 holds for matrix and three-way tensor. Then we will generalize Lemma 1 to all-way tensor.

*Proof.* (Lemma 1 holds for flat 2-LTL matrix) As in figure 1B, we regard flat 2-LTL matrix as a right triangular with two right-angle sides of length $x_1$, $x_2$. The largest rank-1 submatrix is equivalent to the rectangular with sides of length $y_1$ and $y_2$. Let $f(y_1, y_2)$ denotes the area of rectangular. Under geometric constrain, $\frac{y_1}{x_1} = \frac{x_2 - y_2}{x_2}$, s.t., $f(y_1, y_2) = y_1 y_2 = \frac{x_1 y_2 (x_2 - y_2)}{x_2}$. It is clear that $f$ achieves maximum value $f_{max} = \frac{1}{4} x_1 x_2$ when $y_1 = \frac{1}{2} x_1, y_2 = \frac{1}{2} x_2$. As indicated in Figure 1A, the optimal basis (yellow colored) is paralleled with lines (pink colored) anchored on the $1/2$ segmenting point. $\square$

Figure 1: Optimal rank 1 subarray

Figure 2: Suboptimal subarray for $k-1$ LTL tensor

*Proof.* (Lemma 1 holds for 3-way flat 2-LTL tensor) In figure 1D, 3-order flat 2-LTL tensor is depicted as right tetrahedron with three right-angle sides of length $x_1$, $x_2$ and $x_3$, respectively. We also let $f(y_1, y_2, y_3)$ represents the volume of the cuboid of interest. Integrating geometric constrain with Proof 1, $y_1 = \frac{x_1}{2}$, $y_2 = \frac{x_2}{2}$, $\frac{y_1}{x_1} = \frac{y_2}{x_2} = \frac{x_3-y_3}{x_3}$. s.t., $f(y_1, y_2, y_3) = y_3 \cdot \left(\frac{1}{2} \cdot \frac{x_3-y_3}{x_3} \cdot x_1\right) \cdot \left(\frac{1}{2} \cdot \frac{x_3-y_3}{x_3} \cdot x_2\right) = y_3 \cdot \left(\frac{x_3-y_3}{x_3}\right)^2 \cdot \frac{1}{4} x_1 x_2$. When $y_3 = \frac{x_3}{3}$, $f$ get the maximum value $f_{max} = \frac{1}{27} x_1 x_2 x_3$. Additionally, $y_1 = \frac{x_1}{3}$, $y_2 = \frac{x_2}{3}$. As indicated in Figure 1C, the optimal basis (yellow colored) is paralleled with lines (pink colored) anchored on the 1/3 segmenting point. □

*Proof.* (Lemma 1 holds for $k$-order flat 2-LTL tensor, $k > 3$) Since lemma 1 holds for matrix and three-way tensor, the generality of lemma 1 for all-way flat 2-LTL tensor is introduced by mathematical induction. We assume lemma 1 holds for flat 2-LTL tensor with order of $k-1$. And the largest subarray has volume $f_{k-1}(y_1, y_2, ..., y_{k-1})$. For k-way flat 2-LTL tensor, $f(y_1, y_2, ..., y_{k-1}, y_k) = y_k \left(\frac{x_k-y_k}{x_k}\right)^{k-1} f_{k-1}(y_1, y_2, ..., y_{k-1})$, where $f_{max} = \frac{\prod_{i=1}^{k} x_i}{k^k}$ is achieved when $y_k = \frac{x_k}{k}$. By induction, $y_i = \frac{k-1}{k} \cdot \frac{k-2}{k-1} \cdot ... \cdot \frac{1}{2} x_i = \frac{x_i}{k}, \forall i \in [1, k-1]$. In all, Lemma 1 holds for all-way flat 2-LTL tensor. □

**Lemma 2.** *(Geometric perspective in seeding the largest rank-1 pattern) For a $k$ order tensor $\mathcal{X}$ sparse enough and a given tensor size threshold $\lambda$, if its largest rank-1 pattern tensor is larger than $\lambda$, the IRT that reorders $\mathcal{X}$ into a (k-1)-LTL tensor reorders the largest rank-1 pattern to a consecutive block, which maximize the size of the connected solid shape overlapped with the $k-1$ dimension plane over a flat 2-LTL tensor larger than $\lambda$.*

*Proof.* The (k-1)-$LTL$ IRT may reorder the indices of these overlapped patterns to the most bottom left position instead of the largest rank-1 pattern. However, if the tensor is sparse enough, i.e., the overlapped region among rank-1 patterns is relative small, the largest rank-1 patterns will be reordered to form a block in the (k-1)-$LTL$ IRT. In addition, if the size of the overlapped pattern is significant enough, e.g. larger than a given threshold, the overlapped patterns can be identified as a distinct pattern. Otherwise, the largest rank-1 pattern has a distinct solid shape when intersecting with the k-1 dimension plane of the flat 2-$LTL$ tensor that most cross it (Figure 2C,D), while the overlapped patterns in the (k-1)-$LTL$ IRT will intersect with the k-1 dimension plane of the flat 2-$LTL$ tensor most cross it in a ring shape (Figure 2A,B). □

**Lemma 3.** *If a k-order tensor $\mathcal{X}^{m_1 \times \cdots \times m_k}$ can be transformed into a p-LTL tensor by IRT, the p-LTL tensor is unique.*

*Proof.* If the indices of the p-LTL tensor is not unique there are two p-LTL tensor can be achieved by IRT of $\mathcal{X}^{m_1 \times \cdots \times m_k}$, denoted as $\mathcal{X}_{A_1 A_2 ... A_k}$ and $\mathcal{X}_{B_1 B_2 ... B_k}$, where $A_1, ..., A_k$ and $B_1, ..., B_k$

are two permutations of the index sets $\{1, ..., m_1\}, ..., \{1, ..., m_k\}$. Then any p order slice, $X_{\mathbb{P}, I_{\bar{\mathbb{P}}}}$ of $\mathcal{X}_{A_1 A_2, ..., A_k}$ has an identical slice in $\mathcal{X}_{B_1 B_2, ..., B_k}$, which can be denoted as $\mathcal{X}_{\mathbb{P}, I'_{\bar{\mathbb{P}}}}$. By the definition of p-LTL, $\forall j \in \mathbb{P}, 1 \leq j_1 < j_2 \leq m_j$, $T_{sum}(X_{\mathbb{P}, I_{\bar{\mathbb{P}}}}, \mathbb{P} \setminus \{j\})_{j_1} \leq T_{sum}(X_{\mathbb{P}, I_{\bar{\mathbb{P}}}}, \mathbb{P} \setminus \{j\})_{j_2}$ and $T_{sum}(X_{\mathbb{P}, I'_{\bar{\mathbb{P}}}}, \mathbb{P} \setminus \{j\})_{j_1} \leq T_{sum}(X_{\mathbb{P}, I'_{\bar{\mathbb{P}}}}, \mathbb{P} \setminus \{j\})_{j_2}$. Hence, either both $T_{sum}(X_{\mathbb{P}, I'_{\bar{\mathbb{P}}}}, \mathbb{P} \setminus \{j\})_j$ and $T_{sum}(X_{\mathbb{P}, I_{\bar{\mathbb{P}}}}, \mathbb{P} \setminus \{j\})_j$ are identical with respect to $j$, or the index order of the $j$th order are identical in $X_{\mathbb{P}, I_{\bar{\mathbb{P}}}}$ and $X_{\mathbb{P}, I'_{\bar{\mathbb{P}}}}$, suggesting the uniqueness of the p-LTL tensor achieved by IRT of $\mathcal{X}$. $\qquad\square$

**Lemma 4.** *If a k-order tensor is p-LTL, then it is x-LTL, for all the x=p,p+1,...,k.*

*Proof.* For any of its P+1 order slice of $\mathcal{X}$, denoted as $\mathcal{X}_{\mathbb{P}+1, I_{\bar{\mathbb{P}}+1}}$ and $\forall j \in \mathbb{P}+1, 1 \leq j_1 < j_2 \leq m_j$, $T_{sum}(\mathcal{X}_{\mathbb{P}+1, I_{\bar{\mathbb{P}}+1}}, \mathbb{P}+1 \setminus \{j\}) = \sum_{q=1}^{m_t} T_{sum}(\mathcal{X}_{\mathbb{P}+1, I_{\bar{\mathbb{P}}+1}}, \mathbb{P}+1 \setminus \{j, t\})_{:,q}$, where $\mathbb{P}+1$ represents a set of indices with $|\mathbb{P}|+1$ elements, $\mathcal{X}_{\mathbb{P}+1, I_{\bar{\mathbb{P}}+1}}$ is a $|\mathbb{P}|+1$ order slice, and $\{t\} = \mathbb{P}+1 \backslash \mathbb{P}$. Noting $T_{sum}(\mathcal{X}_{\mathbb{P}+1, I_{\bar{\mathbb{P}}+1}}, \mathbb{P}+1 \setminus \{j, t\})$ is a tensor slice sum that takes a $|\mathbb{P}|+1$ order slice as the input and outputs a matrix, which is equivalent to separately compute the tensor slice sum that takes a $|\mathbb{P}|$ order slice with fixed index on the $t$th order as the input and outputs a vector, i.e. $T_{sum}(\mathcal{X}_{\mathbb{P}+1, I_{\bar{\mathbb{P}}+1}}, \mathbb{P}+1 \setminus \{j, t\})_{:,q} = T_{sum}(\mathcal{X}_{\mathbb{P}, I_{\bar{\mathbb{P}}+1} \cup \{i_t = q\}}, \mathbb{P} \setminus \{j\})$. By the definition of $p-LTL$, $T_{sum}(\mathcal{X}_{\mathbb{P}+1, I_{\bar{\mathbb{P}}+1}}, \mathbb{P}+1 \setminus \{j\}, t)_{j_1} = \sum_{q=1}^{m_t} T_{sum}(\mathcal{X}_{\mathbb{P}+1, I_{\bar{\mathbb{P}}+1}}, \mathbb{P}+1 \backslash \{j, t\})_{j_1, q} \leq \sum_{q=1}^{m_t} T_{sum}(\mathcal{X}_{\mathbb{P}+1, I_{\bar{\mathbb{P}}+1}}, \mathbb{P}+1 \backslash \{j, t\})_{j_2, q} = T_{sum}(\mathcal{X}_{\mathbb{P}+1, I_{\bar{\mathbb{P}}+1}}, \mathbb{P}+1 \setminus \{j\}, t)_{j_2}$ $\qquad\square$

## 3 Algorithms

In this sections, we will provide all the pesudo code of algorithms included in GETF framework. Before we illustrate $Pattern\_basis\_finding$ and $Geometric\_folding$, we will introduce some auxiliary algorithms first.

### 3.1 Direction generation

There are in total $k!$ directions to construct a $k-1$ LTL tensor. The first step for GETF is to construct such directions set $\Omega$, where $\mathbf{o} \in \Omega$ is the non repetitive combination of 1 to $k$. Empirically, most of the $k!$ direction will generate the duplicated output resulted from the same or closely related end $k-1$ LTL structural. Such that, normally, $k$ directions are more than enough to generate the suboptimal rank 1 tensor. Still, we provide an $Exha$ Boolean parameter represents exhaustive searching, where $Exha = 1, \Omega' = \Omega$ while $Exha = 0, \Omega'$ is the k sample of $\Omega$. The final $\Omega'$ is the direction set to apply $Geometric\_folding$.

### 3.2 Find the segmenting coordinate

As stated in $Pattern\_fiber\_finding$ algorithm, an essential step is to retrieve the $1/n$ coordinate point. This $POS$ algorithm is designed for this task. The input for $POS$ is a vector $\mathbf{d}$, on which to get the segmenting coordinate $p$, $n$ is the denominator of the segmenting ratio and $s$ is the noise level.

---

**Algorithm 1:** POS

---

**Inputs**: $\mathbf{d}, n, s$
**Outputs**: $p$
$POS(\mathbf{d}, n, s)$:
$\mathbf{m} \leftarrow the\ index\ of\ \mathbf{d} > s$
**if** $length(\boldsymbol{m}) < n$ **then**
$\quad|\quad$ return 0 # no need to segment
**else**
$\quad|\quad q \leftarrow length(\mathbf{m}) // n$
$\quad|\quad \mathbf{m}' \leftarrow order(\mathbf{d_m}, decreasing)$
$\quad|\quad$ return $\mathbf{m'}_q$
**end**

---

## 3.3 Folding tensor based on fiber

This $TENS\_FOLD$ algorithm allows an efficient folding of $k$ order tensor into $k-1$ order tensor by one dimension on the basis of a specific fiber as mentioned in the main content.

---

**Algorithm 2:** TENS_FOLD

---

**Inputs**: a $k$-order tensor $\mathcal{X}^{m_1 \times m_2 \cdots \times m_k}$, the fiber $\mathbf{f}$ to be fold upon, and directions of this fiber $\mathbf{o}$
**Outputs:** the $(k-1)$ order tensor $\mathcal{H}$
$TENS\_FOLD(\mathcal{X}, \mathbf{o}, \mathbf{f})$:
$d \leftarrow diff(range(m), \mathbf{o})$ #the fold dimension
$\mathcal{H} \in \{0\}^{m_1 \times \ldots \times m_{d-1} \times m_{d+1} \ldots \times m_k}$ # initialization
$\mathcal{H}_{i_1 i_2 \ldots i_{d-1} i_{d+1} \ldots i_k} \leftarrow \mathcal{X}_{i_1 i_2 \ldots i_{d-1} : i_{d+1} \ldots i_k} \cdot \mathbf{f}$
return $\mathcal{H}$

---

## 3.4 2 LTL projection

From Lemma 1, the $1/k$ segmentation pinpoint the optimal pattern basis for the $flat\ 2-LTL\ tensor$. We could easily derive the (k-1)-$LTL$ throught IRT. However, owing to the impact of different levels of noise, lemma 1 does not hold on $(k-1)-LTL$ tensor in definite. The 2_LTL_projection algorithm is to fill this gap. 2_LTL_projection finds a proper searching space that maintains the $flat\ 2-LTL\ tensor$ property with limited noise. With 2_LTL_projection, we could find the biggest rank 1 tensor patterns in the first iteration, the second biggest patterns in the second iteration, so on and so forth. In practise, we omitted this process to further accelerate GETF. Without the 2_LTL_projection, the order of pattern size may vary with its sequential order. But it does not impact the overall decomposition efficiency. A similar study on Boolean matrix can be found in [1].

---

**Algorithm 3:** 2_LTL_projection

---

**Inputs:** a $k$-order K-1 LTL tensor $\mathcal{X} \in \{0,1\}^{m_1 \times m_2 \cdots \times m_k}$ and the range of flat 2-$LTL$ tensor
$\quad \{m_1^l, ..., m_1^h\} \otimes, ..., \otimes \{m_k^l, ..., m_k^h\}$
**Outputs:** the flat 2-$LTL$ tensor maximizes the overlap between its k-1 dimension plane and $\mathcal{X}$
$S \leftarrow 0^{(m_1^h - m_1^1 + 1) \times \ldots \times (m_k^h - m_k^1 + 1)}$
**for** $(i_1, i_2, \ldots i_k) \in \{1, ..., m_1^h - m_1^l + 1\} \otimes, ..., \otimes \{1, ..., m_k^h - m_k^l + 1\}$ **do**
$\quad \mathcal{X}^{projected} \leftarrow \text{project}(\mathcal{X}, Plane(i_1, i_2, \ldots i_k))$
$\quad S_{i_1, i_2, \ldots i_k} \leftarrow sum(neighbor\_weighted\_scoring(\mathcal{X}^{projected}))$
**end**
$i_1^*, i_2^*, \ldots i_k^* \leftarrow argmax\{S\}$
return$(i_1^*, i_2^*, \ldots i_k^*)$

---

## 3.5 Pattern fiber finding

Assume we find the pattern fiber along the direction $\mathbf{o} \in \{1, ...k\}$. Derive from lemma 1. The candidate pattern fiber is revealed recursively. The recursive algorithm first computes tensor slice sum of $\mathcal{X}$ through mode $\mathbf{o}_1$ to $\mathbf{o}_{k-1}$, i.e., from $T_{sum}(\mathcal{X}, \{\mathbf{o}_1\})$ to $T_{sum}(\mathcal{X}, \{\mathbf{o}_1, ..., \mathbf{o}_{k-1}\})$ and identify the first coordinate on the mode $\mathbf{o}_k$ as the $1/k$ segmenting point of the computed slice sum. Then in each recursive iteration n, there are $k-1$ computed coordinates for the mode of $\mathbf{o}_{k-n+1}$ to $\mathbf{o}_k$ and the $k-n$ order tensor of the slice sum of $\mathcal{X}$ through mode $\mathbf{o}_1$ to $\mathbf{o}_{k-n}$ computed previously, which together form a vector, on which the $1/(n+1)$ segmenting point is computed as the coordinate for the mode $\mathbf{o}_{k-n}$. Denote $(i_1^0, ..., i_{\mathbf{o}_1-1}^0, i_{\mathbf{o}_1+1}^0, ..., i_k^0)$ as the identified coordinates, the mode-$\mathbf{o}_1$ pattern fiber $\mathbf{a}^{m_{\mathbf{o}_1} \times 1, \mathbf{o}_1} = \mathcal{X}_{i_1^0 \ldots i_{\mathbf{o}_1-1}^0 i_{\mathbf{o}_1+1}^0, \ldots i_k^0}$. Detailed algorithm implementation is as followed.

The recursive algorithm first computes the tensor slice sum of $\mathcal{X}$ through mode $\{\mathbf{o}_1\}$ to $\{\mathbf{o}_{k-1}\}$ and identify the first coordinate on the mode $\mathbf{o}_k$ as the $1/k$ segmenting point of the computed tensor slice sum. Then in each recursive iteration n, there are $k-1$ computed coordinates for the mode of $\mathbf{o}_{k-n+1}$ to $\mathbf{o}_k$ and the $k-n$ order tensor of the slice sum of $\mathcal{X}$ through mode $\mathbf{o}_1$ to $\mathbf{o}_{k-n}$ that computed previously, which together form a vector, on which the $1/(n+1)$ segmenting point is

---

**Algorithm 4:** Pattern_fiber_finding

---

**Inputs:** a $k$-order tensor $\mathcal{X} \in \{0,1\}^{m_1 \times m_2 \dots \times m_k}$, the finding direction $\mathbf{o}$ as defined in algorithm 1, a segmentatin denominator $n$ and the noise control prarmeter $s$.

**Outputs:** the coordinates $(i_1, ..., i_{\mathbf{o}_1-1}, i_{\mathbf{o}_1+1}, ..., i_k)$ of the mode-$\mathbf{o}_1$ basis fiber.

$Pattern\_fiber\_finding(\mathcal{X}, \mathbf{o}, s)$ :

**if** *is.vector($\mathcal{X}$)* **then**
> return $POS(\mathcal{X}, n+1, s)$

**else**
> $n \leftarrow n+1$ #total order of the current $\mathcal{X}$
> $\mathcal{X} \leftarrow TENS\_FOLD(\mathcal{X}, \mathbf{o}_n)$ #compute the $T_{sum}(\mathcal{X}, \{\mathbf{o}_n\})$ for the current $\mathbf{o}_n$ mode.
> $Cor \leftarrow Pattern\_fiber\_finding(\mathcal{X}, \mathbf{o}, s)$ # recursion starts for folding next dimension and return coordinates.
> $i_{\mathbf{o}_n-n} \leftarrow POS(\mathcal{X}_{:...:i_{\mathbf{o}_n-n+1}:...:i_{\mathbf{o}_n}:...:, n+1, s})*$
> $Cor \leftarrow append(Cor, i_{\mathbf{o}_n-n})$ #integrate coordinates information
> return $Cor$

**end**

*$i_{\mathbf{o}_n-n+1}, ..., i_{\mathbf{o}_n}$ are currently computed $n$ coordinates of the mode $\mathbf{o}_n - n + 1$ to $\mathbf{o}_k$ of the pattern fiber.

---

computed as the coordinate for the mode $\mathbf{o}_{k-n}$. Denote $(i_1^0, ..., i_{\mathbf{o}_1-1}^0, i_{\mathbf{o}_1+1}^0, ..., i_k^0)$ as the identified coordinates, the mode-$\mathbf{o}_1$ pattern fiber $\mathbf{a}^{m_{\mathbf{o}_1} \times 1, \mathbf{o}_1} = \mathcal{X}_{i_1^0...i_{\mathbf{o}_1-1}^0 i_{\mathbf{o}_1+1}^0...i_k^0}$.

Owing to the recursive property of Pattern_fiber_finding, for the $i$th iteration of a tensor has the size of $m^k$, the computational cost is $m^i$ for $TENS\_FOLD$ and $mlog(m)$ for $POS$. Such that, the overall computational cost for Pattern_fiber_finding is $\sum_{i=1}^{k} m^i + mlog(m) = \frac{m^{k+1}-m}{m-1} + kmlog(m)$.

## 3.6 Geometric folding

The geometric folding approach is to compute the rank-1 tensor component best fit via $\mathcal{X}$ from the pattern fiber identified by the $Pattern\_fiber\_finding$ algorithm. For a $k$-way tensor $\mathcal{X}^{m_1 \times m_2 \dots \times m_k}$ and a pattern fiber identified by $Pattern\_fiber\_finding$, denoted as $\mathcal{X}_{:i_2^0...i_k^0}$. The algorithm further computes the inner product of between $\mathcal{X}_{:i_2^0...i_k^0}$ and each fiber $\mathcal{X}_{:i_2...i_k}$, where $i_2 = 1...m_2$, ..., $i_k = 1...m_k$, which generates a new $k-1$ order tensor $\mathcal{H}^{m_2 \times m_3 \dots \times m_k}$ and $\mathcal{H}_{i_2 i_3...i_m} = \sum_{j=1}^{m_1} \mathcal{X}_{ji_2^0...i_k^0} \wedge \mathcal{X}_{ji_2...i_k}$. This new tensor is further discretized based on a user defined noise tolerance level and generates a new binary $k-1$ order tensor $\mathcal{X}'^{m_2 \times m_3 \dots \times m_k}$. We call this approach as geometric folding of a $k$-order tensor into a $k$-1 order tensor based on the pattern fiber $\mathcal{X}_{:i_2^0...i_k^0}$. Then a pattern fiber of $\mathcal{X}'^{m_2 \times m_3 \dots \times m_k}$ will be identified by $Pattern\_fiber\_finding$, based on which $\mathcal{X}'^{m_2 \times m_3 \dots \times m_k}$ will be further folded into a $k-2$ order tensor. The algorithm will be executed to fold the $k$-way tensor into a 2 dimensional matrix with $k$-2 rounds of the pattern fiber finding and geometric folding, which will generate $k$-2 pattern fibers. The pattern fibers of the last 2 dimensional will be further derived via a discretization of the folded matrix and a BMF by using MEBF [1]. It is worth to note that the $Pattern\_fiber\_finding$ follows the same direction as the further geometric folding process.

$Geometric\_folding$ also follows the recursive computing structure as $Pattern\_fiber\_finding$. For $i$th iteration, the computation is $\frac{m^{i+1}-m}{m-1} + imlog(m)$ for $Pattern\_fiber\_finding$, $m^i$ for calculating the inner product and $m^{i-1}$ for the discretization. Such that, the total computational cost for $Geometric\_folding$ is $\sum_{i=1}^{k} \frac{m^{i+1}-m}{m-1} + imlog(m) + m^i + m^{i-1} = \frac{2m^{k+2}-m^k}{(m-1)^2} + \frac{1-2m^2}{(m-1)^2} - \frac{km}{m-1} + \frac{k(k+1)}{2}mlog(m)$.

## 3.7 Noise control

GETF tackles Boolean tensor decomposition in a geometric view. However, randomized noise will hurt the geometric property which will result in identifying less representative pattern fibers. In figure 3, we illustrate this situation in a 2D data. If we do not consider the noise, the segmenting

**Algorithm 5:** Geometric_folding

**Inputs:** A $k$-order tensor $\mathcal{X}^{m_1 \times m_2 \ldots \times m_k}$, the direction of pettern fiber finding and geometric folding $\mathbf{o}$ as defined in algorithm 1 and a noise tolerance parameter $t$

**Outputs:** Bases of the rank-1 tensor component $\mathbf{a}^1, \mathbf{a}^2, ..., \mathbf{a}^k$

$Geometric\_folding(\mathcal{X}, \mathbf{o}, t):$

$\mathcal{X}^{original} \leftarrow \mathcal{X}$

$s \leftarrow NOISE\_CONTROL(\mathcal{X})$ # detect the noise level in $\mathcal{X}$

$\mathbf{o}^{original} \leftarrow \mathbf{o}$

**for** $i = 1, ..., k-2$ **do**

$\quad \mathbf{o} \leftarrow \mathbf{o}^{original}_{i...k}$

$\quad |\mathbf{o}| = k - i + 1, \mathbf{o}_j = \mathbf{o}_j - 1, \, if \, \mathbf{o}_j > \mathbf{o}_{i-1}$

$\quad (i^0_1, ..., i^0_{\mathbf{o}_1-1}, i^0_{\mathbf{o}_1+1}, ..., i^0_{|\mathbf{o}|}) \leftarrow Pattern\_fiber\_finding(\mathcal{X}, \mathbf{o}, s)$

$\quad \mathbf{a}^{m_{\mathbf{o}_1} \times 1, \mathbf{o}_1} = \mathcal{X}_{i^0_1 \ldots i^0_{\mathbf{o}_1-1} i^0_{\mathbf{o}_1+1} \ldots i^0_{|\mathbf{o}|}}$

$\quad \mathcal{H}_{i_1 i_2 \ldots i_{|\mathbf{o}|}} \leftarrow \sum_{j=1}^{m_{\mathbf{o}_1}} \mathcal{X}_{i^0_1 \ldots i^0_{\mathbf{o}_1-1} i^0_{\mathbf{o}_1+1} \ldots i^0_{|\mathbf{o}|}} \wedge \mathcal{X}_{i_1 \ldots i_{\mathbf{o}_1-1} i_{\mathbf{o}_1+1} \ldots i_{|\mathbf{o}|}}$

$\quad \mathcal{X} \leftarrow \mathcal{H} \cdot 0$

$\quad \mathcal{X}_{i_1 \ldots i_{\mathbf{o}_1-1} i_{\mathbf{o}_1+1} \ldots i_{|\mathbf{o}|}} \leftarrow 1 \text{ if } \mathcal{H}_{i_1 \ldots i_{\mathbf{o}_1-1} i_{\mathbf{o}_1+1} \ldots i_{|\mathbf{o}|}} \geq t \cdot |\mathbf{a}|$

**end**

$(\mathbf{a}^{m_{\mathbf{o}_{k-1}} \times 1, \mathbf{o}_{k-1}}, \mathbf{a}^{m_{\mathbf{o}_k} \times 1, \mathbf{o}_k}) \leftarrow MEBF(X, t, s)$

$\mathbf{a}^{\mathbf{o}_i} \leftarrow \mathbf{a}^{m_{\mathbf{o}_i} \times 1, \mathbf{o}_i}, i = 1, ..., k$

return $\mathbf{a}^1, ..., \mathbf{a}^k$

Figure 3: Noise control illustration of GETF

process will get the misleading patterns (yellow boxed in figure 3A). To identify the level of noise, we calculate the slice sum of this matrix on either of its two directions. And fit two Gaussian distribution corresponds to two modes based on the slice sum (figure 3C). The underlying assumption is that noise or zero will dominate the left mode while the right mode is responsible for actual patterns. Also, the right mode is not aiming to present actual pattern in exact, but to distinguish the noise level. The noise level is regarded at the count number where p-value is 0.9, indicated by the dashed red line in figure 3C. After this noise detection level, the adjusted fiber base (yellow boxed in figure 3B) regains its representation and thus recovered the desired rank 1 pattern.

### 3.8 Method discussion

**Lemma 1**, **3**, and **4** are mathematically rigorous while **Lemma 2** is relatively descriptive due to the errors and level of overlaps among pattern tensors cannot be generally formulated, especially in a high order tensor. However, our derivations in APPENDIX reflects the geometric property described in **Lemma 2** stands for most of the tensors whose pattern tensors are not heavily overlapped. The advantage of GETF is significant. The computational cost of the IRT and identification of the flat $2$-$LTL$ tensor mostly cross the largest pattern are all $O(n)$, where $n$ is the tensor size. The property of the position of the most likely pattern fiber enables circumventing heuristic greedy search or optimization for seeding the largest rank-1 pattern. Due to the heuristic consideration of the algorithm, we focused on the method performance and robustness evaluation on an extensive set of synthetic data to demonstrate GETF is robust for high order tensor decomposition with different level of overlapped patterns and errors, followed by the applications on real-world datasets.

# 4 Experimental Results on Synthetic Datasets

In this section, we highlight the performance comparison of GETF with state-of-the-art approaches, The evaluation focus on two metrics, time consumption and reconstruction error, defined below [2, 3]. Denote $A^{m_1 \times l,1}, A^{m_2 \times l,2}, ..., A^{m_k \times l,k}$ is the true pattern matrices of a $k$-order tensor $A^{m_1 \times l,1} \otimes A^{m_2 \times l,2} ... \otimes A^{m_k \times l,k}$, and $A^{*m_1 \times l,1}, A^{*m_2 \times l,2}, ..., A^{*m_k \times l,k}$ as the pattern matrices identified by each algorithm. All the experiments is running on the same super computer with 1 thread and 32G memory.

$$Reconstruct\ error :=$$
$$\frac{(A^{m_1 \times l,1} \otimes ... \otimes A^{m_k \times l,k}) \ominus (A^{*m_1 \times l,1} \otimes ... \otimes A^{*m_k \times l,k})}{\prod_{i=1}^{k} m_i}$$

Binary tensor is generated at the following scheme.

$$\mathcal{X}^{m_1 \times m_2 ... \times m_k} = A^{m_1 \times l,1} \otimes A^{m_2 \times l,2} ... \otimes A^{m_k \times l,k} + \mathcal{E}_f$$

Where

$$A_{ij}^{m_1 \times l,1}, A_{ij}^{m_2 \times l,2}, ..., A_{ij}^{m_k \times l,k} \sim Bernoulli(p_0)$$

$$\mathcal{E}_{f_{i_1 i_2 ... i_k}} \sim Bernoulli(p)$$

"$\mathcal{E}_f$" is a flipping operation which introduce errors. Overall

$$\mathcal{X}_{i_1 i_2 ... i_k} = \begin{cases} \vee_{j=1}^{l}(A_{i_1 j}^{m_1 \times l,1} \wedge ... \wedge A_{i_k j}^{m_k \times l,k}), \mathcal{E}_{f_{i_1 i_2 ... i_k}} = 0 \\ \neg \vee_{j=1}^{l}(A_{i_1 j}^{m_1 \times l,1} \wedge ... \wedge A_{i_k j}^{m_k \times l,k}, \mathcal{E}_{f_{i_1 i_2 ... i_k}} = 1) \end{cases}$$

To comprehensively benchmark GETF, we conducted the method evaluation on 4 scenarios, namely (1) low density tensor without error, (2) low density tensor with error, (3) high density tensor without error and (4) high density tensor without error. The density level is defined as $d = \frac{|\mathcal{X}|}{\prod_{i=1}^{k} m_i}$. We set $d = 0.1$ and $d = 0.3$ for low and high density, respectively. The number of rank-1 tensor component l was set as 5 for all simulated data. The density level was achieved by adjusting $p_o$. The error level p was set as 0.01. 20 simulations were conducted for each scenario. Averaged running time and reconstruction error rate of 20 simulations were used for method comparison.

It is noteworthy that the GETF is capable for decomposing $k$-order binary tensor for $k \geq 2$ while most of other methods were designed for one specific $k$. Based on the recent reviews, we select to benchmark GETF with MP on the 2D BMF problem and with LOM on the 3D BTD problem [2, 4]. Both of the selected methods were recently developed and achieved high performance when comparing with other methods. The convergent condition $\tau$ for all the algorithms is set as either of the following criterion is met: 1) 10 rank-1 tensor component were identified; 2) the cost function $\gamma$ was not decreasing with newly identified pattern. It is noteworthy that increasing number of patterns is equivalent to increasing the number of latent layers for fitting the probability distribution in the LOM methods [4].

Figure 4 presented the reconstruction error (y-axis) of GETF, MP and LOM algorithms as a function of the number of components on the x-axis, for different scenarios: low/high signal density, with/without error, tensor dimensions. Standard derivation of the reconstruction error on 20 replicates is very low for all algorithms, suggesting a robust performance, and it is represented by the size of the dot shapes. GETF largely outperformed the other methods for all the scenarios except for the high density with error case, where both methods performed comparatively. Notably, the reconstruction error curve of GETF flattened after reaching the true number of components, i.e., 5, suggesting the robustness to noisy of GETF. Figure 5 shows the time consumption of GETF, MP and LOM, where the error bar stands for standard derivation for 20 replicates. Apparently, GETF is in orders of magnitude faster than MP and LOM. In summary, GETF is the fastest and most robust algorithm compared with MP and LOM across different scenarios. GETF achieves favorable reconstruction error with much less time consumption than other algorithms. MP suffers in decomposing both low density data and noisy data, and its convergence rate is slow compared with GETF. LOM performed robustly on noisy data, but being a probability fitting model, LOM is severely impacted by the density of the data.

We show the performance of GETF on high order tensor data in Figure 6. Noted, when the order increases, the tensor size would increase exponentially. BTD is very likely to become a memory

Figure 4: Performance of GETF, MP and LOM on the accuracy of decomposition

Figure 5: Performance of GETF, MP and LOM on time consumption

Figure 6: Performance of GETF on high order tensor data

Figure 7: Statistics of Chicago crime data analysis

Figure 8: The map visualization of Chicago city crime regions

bounded problem. In this case, an O(n) algorithm like GETF is suitable to tackle such scenarios. GETF showed consistent performance with or without noise. Most importantly, for a 5-way tensor with around $3*10^8$ elements, GETF finished decomposition in 1 minutes. Overall, the simulated results truly advocated the efficiency and robustness of GETF for Boolean tensor decomposition, irrespective of tensor order, data size, data density and noise level.

# 5 Experimental Results on Real Data

We also benchmarked GETF on two real-world data sets. Specially, the reconstruct error in the real world experiments is defined by:

$$Reconstruct\ error := \frac{|\mathcal{X} \ominus (A^{1*} \otimes A^{2*}... \otimes A^{k*})|}{\prod_{i=1}^{k} m_i}$$

, where $\mathcal{X} \in \mathbb{R}^{m_1 \times m_2... \times m_k}$ denotes a real-world tensor data, and $A^{1*}...A^{k*}$ represent the identified bases for each k orders.

## 5.1 Chicago crime data

A real world event usually concerns multiple properties. Take crime as an example, when and where it happens, whether arrested or not, which type it is, are all important properties of a crime event. Here we take Chicago crime data[1] as an example to benchmark GETF performance and illustrate its applications of high-order binary data in a complex event setting.

Located on the shore of Lake Michigan, the windy city Chicago is the most populous city in the Midwest US. High population boosts economy but also breeds crimes. Unfortunately, being the central hub for US, Chicago suffered the highest crime rate across nation. Here we apply GETF to analyze Chicago crime data from 2001 to 2019. The reasons we choose this dataset are as follows, 1) the commitment of crimes always have reasons or traces, which means, in general, crime data contains certain pattern information. For example, most crimes such as theft and robbery have strong date patterns. Criminals committed such action are frequently triggered by the need to pay rent or credit cards that has a strict deadline once per month. In [1], we analyzed the relation of crime commitment with date for individual years. By applying the all-way BTD method GETF, we could

simultaneously mine the relations of crimes with date, year, arrest, domestic et al in a general and comprehensive manner.

Here we construct a 3D tensor data from the crime records and evaluate performances of GETF with LOM. We divide Chicago city area into 436 subregions of roughly equal size. The tensor is thus constructed by three dimensions, subregions, date and year $\mathcal{X}^{m_1 \times m_2 \times m_3}$. Where $m_1$ is equal to 436, $m_2$ is 365 means the common days in a year. And $m_3$ is 19 representing 19 year of crime records. $\mathcal{X}_{i_1 i_2 i_3} = 1$ indicates $i_1$ region has non arrested crimes reported on the $i_2$ date in $i_3$ year. GETF and LOM was then applied on the construct binary tensor data to retrieve $A^{1*}$, $A^{2*}$ and $A^{3*}$. The convergence criteria $\tau$ is the same as in simulation comparison, 1) 10 patterns has been identified, 2) the cost function stopped decreasing with newly identified patterns. Figure 8A shows the changes of GETF and LOM in reconstruct error along with the addition of pattern numbers. GETF showed clear advantage over LOM with faster decline in reconstruction error. GETF plateured after the first two patterns, while it is more than eight for LOM (Figure 7A), which indicates GETF retrieved patterns are of significantly importance that conveyed dominant information of the original data. Next, we investigate the top GETF identified patterns to validate its capacity in the application of data mining.

To introduce the complex data scenario, we construct a 4D tensor data based on the crime report with dimensions as subregion, year, data and arrest. Here arrest can be regarded as a representative factor of the severeness of the crime. The tensor data is generalized as $\mathcal{H} \in \mathbb{R}^{436 \times 365 \times 19 \times 2}$, where $\mathcal{H}_{i_1 i_2 i_3 1} = s$ indicates $i_1$ region has $s$ non-arrested crimes reported on the $i_2$ date in $i_3$ year and $\mathcal{H}_{i_1 i_2 i_3 2} = s$ indicates $i_1$ region has $s$ arrested crimes reported on the $i_2$ date in $i_3$ year. The Boolean relation tensor data $\mathcal{X}$ is thus derived from $\mathcal{H}$. I.e., $\mathcal{X}_{i_1 i_2 i_3 i_4} = 1 \; if \; \mathcal{H}_{i_1 i_2 i_3 i_4} > 0$. GETF is currently the only tool to handle such higher order Boolean tensor data $(k = 4)$, where GETF decompose original data $\mathcal{X}$ into $\hat{\mathcal{X}} = A^{436 \times l, 1*} \otimes A^{365 \times l, 2*} \otimes A^{19 \times l, 3*} \otimes A^{2 \times l, 4*}$. To help illustrate the finding of GETF, we introduce follows metrics specific to this data scenario.

$$Crime\ index := \sum_{i_2=1}^{365} \sum_{i_4=1}^{\bar{2}} \hat{\mathcal{X}}_{i_1 i_2 i_3 i_4}$$

$$Crime\ Dates := \sum_{i_2=1}^{365} \sum_{i_4=1}^{\bar{2}} \mathcal{X}_{i_1 i_2 i_3 i_4}$$

$$Crime\ Counts := \sum_{i_2=1}^{365} \sum_{i_3=1}^{19} \sum_{i_4=1}^{2} \mathcal{H}_{i_1 i_2 i_3 i_4}$$

A very import metric to evaluate the crime situation for a specific region is to see how many days in a year on average when crime has been committed. And by decomposition, the reconstructed tenor $\hat{\mathcal{X}}$ conveys the date pattern information. Thus, Crime index is the yearly mean value of number of crimes dates in reconstruct tensor data, which gives a distinctive overall evaluation of the crime situation in that region.

Unlike crime index defined on the reconstructed tensor, Crime dates is defined on the original tensor date. I.e. data pattern has not been refined or mined between features like crimes, region, dates et al.

Moreover, to evaluate the identified patterns can truly cover information. Here we introduce a outsider factor . This factor is the total number of crimes happened in that region. This factor alone does not convey any other information like date, year or arrest, which could serve as the overall indicator of our observed patterns.

We take the first two patterns from GETF decomposition in reconstruct our tensor data as they contains the most of the information. Figure 7B,C,D represent the Crime index, Crime Dates and Crime counts for 436 regions, respectively. In figure 7B, we witness a clear difference of regions on Crime index. Regions marked with red dots has year round crime rate even in reconstructed tensor. But regions displayed as blue dots has no date related crime patterns, indicates a rather safe environment. And some regions marked with gray dots is slightly better than red dots regions. All these regions have very noisy distribution on Crime Dates (figure 7C). Some blue regions that are identified as safe region showed year round Crime Dates. Here, we introduced the Crime counts metric in figure 7D. We deem this metric does not relate to dates, year or arrest. Such that, it could

Figure 9: Spatial tensor patterns of breast cancer tissue

Figure 10: Analysis of breast cancer tensor pattern

reflect the crime situation of blue or red regions. In figure 7D, we witness a very clear distinction between blue and red regions. All the blue regions are underneath the red regions showing a decrease crime numbers. This justified the data mining approach of identify different regions by GETF. The blue(safe) region and red(dangerous) region, are refined from Date-related tensor data but shows the general representing for the overall crime situation indicated by crime counts in figure 7D. Noted, this classification can not be easily achieved by looking into Crime Dates directly, as indicated by figure 7C. We also found that the Arrest factor does not make much difference in determine the blue or red region. The explanation maybe that the severeness of a crime does not related to the frequency of crimes. In figure 8, we visualize blue and red region on the map level. These regions also showed geometric patterns. Blue regions reside at peripheral areas of the city. While red regions lie in the center.

This example illustrates the ability of GETF in mining relational date with multiple properties, aka, higher order binary tensor data. As suggested, GETF derived patterns represent the true property of subregions, which is hard to derive by simply analyzing the original data. This example also displayed the application of BTD methods in classification problems, as GETF accurately classified safe and dangerous regions.

## 5.2 Spatial transcriptomics

Spatial molecular data is another example of high order tensor data. 3-D spatial transcriptomics data composes four orders, namely three spatial coordinates X-Y-Z and the gene features. Here we benchmark GETF on a breast cancer 3-D spatial transcriptomics data[2] and highlight the application of GETF in this data scenario that are with high noise level.

Recent single cell RNA sequencing technologies enable research to observe expression level of more than 20,000 gene features in single cell resolution [5, 6], among which spatial single cell RNA-

sequencing data is one popular data type that can bridge molecular data with tissue spatial histological information. Associating molecular features with spatial information on single cell resolution grants a capability to explore of cell-cell interactions and cellular level signaling in complex tissues and disease such as brain tissue and solid cancer, et al [7, 8, 9, 10]. A typical spatial transcriptomics data is of 4 orders, i.e., gene features and X-Y-Z of 3D spatial coordinates. Here we apply GETF method on a breast cancer spatial data set [7] consists of 1020 cell positions ($x \times y \times z = 15 \times 17 \times 4$) with 13360 genes features, i.e., $\mathcal{H} \in \mathbb{R}^{13360 \times 15 \times 17 \times 4}$.

The 3-D schematic diagram of a breast cancer tissue and the distribution of breast cancer cells with certain gene expression patterns are illustrated in figure 9 [11]. The data conceived the information of how the cells of different types and gene expressions distribute in a cancer tissue. Two distinct types of cell distributions, namely (1) sparsely distributed cells of a rank-1 pattern (red blocks in figure 9) and (2) densely distributed rank-1 patterns (blue block in figure 9). Here we apply GETF to identify the cells with a certain number of gene features and spatial coordinates that form a rank-1 pattern. The cells with the two types of distributions should be distinct biological characteristics, especially in cell-cell interaction mechanism, since the 1st type of distribution have cells interact with other cell types with cell of the 2nd type of distribution tend to only interact with themselves.

We first discretized the observed spatial transcriptomics data and build it into a 4D tensor $\mathcal{X} \in \{0, 1\}^{13360 \times 15 \times 17 \times 4}$, by $\mathcal{X}_{ijkl} = 1 \ if \ \mathcal{H}_{ijkl} > 0$ and $\mathcal{X}_{ijkl} = 0 \ if \ \mathcal{H}_{ijkl} = 0$.Noted, unlike the Chicago crime data, spatial transcriptomics data is genuinely noisy due to multiple sources of experiment error. For GETF, $\tau$ was set to get at most 10 patterns or the cost function stopped decreasing, the same as above content. Noise tolerance parameter $t$ was set to 0.6 to cope with the high noisy data scenario. LOM was conducted by using default parameters. We first compared the performance of GETF and LOM on the 3D slices of this data, $\mathcal{X}_{:::l} \in \{0, 1\}^{13360 \times 15 \times 17}$ for different l. Figure 10A shows the reconstruct error of both methods along with the addition of pattern number. The LOM method failed to identify any rank-1 pattern from this data set. On the other hand, GETF managed to derive patterns gradually, which indicates the versatility of GETF in both clean (Chicago crime) and noisy (spatial transcriptomics) real world data.

We also applied GETF to the whole 4-D tensor to identify the rank-1 patterns with sparse and dense distribution, where the sparseness is determined by a parameter $s$ defined by the number of cell positions divided by the volume of the smallest cuboid that covers all the positions. The bigger the sparsity parameter $s$, the denser the pattern. Among the top 10 patterns, we analyzed the two extremest patterns with $s = 0.5$ (red) and $s = 1.0$ (blue), respectively. The sparse pattern has 24 cell position all express 232 genes ($232 \times 4 \times 4 \times 2$). The dense pattern has 90 cell positions all express 40 genes ($40 \times 15 \times 3 \times 2$). We first obtained the general information of these cell positions by visualizing them in lower dimensions using UMAP[12]. We observed clear difference between sparse positions (red dots) and dense positions (blue dots) (figure 10B).

We further justified the top rank-1 patterns with sparse and dense spatial distributions identified by GETF with their biological significance. In order to explore how the cells with the two distribution types behave on individual gene level, we conduct gene ontology enrichment analysis of the gene features of the sparse and dense patterns. We showed the top 5 biological processes enriched by the genes of each pattern in figure 10C and 10D, where each bar illustrates the negative logarithm of the P value of each pathway assessed by hypergeometric test. Cells of the dense pattern positions are with high expression of the genes related to translation (ribosome small subunit assembly, translational initiation, SRP-dependent cotranslational protein targeting to membrane) and transcription (viral transcription, nuclear-transcribed mRNA catabolic process, nonsense-mediated decay) (figure 10C). Whereas the cell positions of sparse pattern indicate the properties of cancer peripheral tissues. These positions are in charge of maintaining cancer tissue structure (positive regulation of amyloid fibril formation, positive regulation of intracellular mRNA localization), stabilizing cancer cells (telomerase holoenzyme complex assembly), battling immune system (astrocyre activation involved in immune response) and suppressing anti-tumor effect (negative regulation of polynucleotide adenylytransferase) (figure 10D). Biological implication can be made here includes the dense pattern keeps the vibrancy of cancer tissue by showing strong activities in transcription and translation, while the sparse pattern maintains the tissue structure and suppress anti-tumor immune effect. Our analysis demonstrated that the GETF-derived patterns reveal the complicated but integrated spatial structure of breast cancer tissue with different functionalities.

In summary, we validated the performance and application of GETF in these two real-world data. To the best of our knowledge, GETF is the first all-way Boolean tensor factorization method, which brings vast potential in analyzing relation events in a complex setting like spatial-temporal crime records data and spatial biological data. In addition, our analysis demonstrated the GETF derived patterns accurately represent the low rank structure of the data. In certain case, GETF also serve as a classification approach that could not be easily achieved by analyzing original data.

## Footnotes

[1]Chicago crime records downloaded on March 1st, 2020 from https://data.cityofchicago.org/Public-Safety

[2]Breast cancer spatial transcriptomics data is retrieved from https://www.spatialresearch.org/resources-published-datasets/doi-10-1126science-aaf2403/