[Reviews · NeurIPS 2020]

Review 1

Summary and Contributions: The authors propose a Boolean Tensor Decomposition technique called GETF. GETF performed various experiments on both synthetic and real datasets including crime data in Chicago and breast cancer data. Related work and problem definition sections are clear, however, experiments have fundamental issues.

Strengths: N/A

Weaknesses: There exist several limitations in experiments which are summarized as follows: 1-Authors need to show useful scenarios where binary representation is the best way to model the tensor.  For instance, it seems count representation makes more sense rather than the binary representation. By binary representation, the quantity of crimes is ignored. att least authors need to verbally mention why binary representation is better than count representation. 2-The author mentioned GETF is more scalable and faster than other baselines, however, scalability experiments need more work. For figures 6c, 6I, it is better to specify the number of patterns. Why not showing the scalability of GETF and other baselines as we increase 1- number of patterns 2- the size of a tensor dimension, and 3- number of non-zero elements in the tenor.  Moreover, it is very important to compare the scalability of GETF and other baselines on real-world datasets including Chicago crime record, and breast cancer data sets. Currently, the scalability experiment on real-world data is missing. Another minor issue is that the authors mentioned "We also evaluated each algorithm on different data scale in supplementary materials". It is better and easier for the reader to specify the section number. 3- Authors mentioned, "For a 5-way tensor with more than 3 ∗ 108 elements, GETF completed the task in less than 1 minute." This sentence is very vague.  It is not clear what "completed the task" means? What is the convergence tolerance?  what is the number of patterns? Is it 5? The authors need to mention that. 4- If I understand correctly, I assume the y-axis in Figures Figure 7C and 7D is the same, however, one is crime index and the other one is crime data. Moreover, it is necessary to mention why GETF is able to distinguish the two regions better than other baselines. Also, it is not clear how the authors determine the crime index for each region. Minor issues: 1-There is no description of figure 7A.   2- The caption of Figure 8 is vague. Does the visualization come from GETF? or is it ground truth representation.   2-Authors need to provide more details about the captions in Figures 6 and 7. 3-I suggest authors provide the statistics of the data sets in a table. 4- Line 274: data densitie and noise level ->data densities and noise levels. 5- "On the other hand, GETF manages to derive patterns gradually (Figure 7I)". Decreasing in reconstruction error doesn't neeccesarly mean that a technique is finding patterns. 6- line 225, 226: "Figure 8A shows the changes of  GETF and LOM in reconstructing error along with the addition of pattern numbers". There is no figure 8a. 7- It would be better to include the code in the submission. 8- Provide the notations and symbols in a table. 

Correctness: Please check the Weaknesses section.

Clarity: No, it is very hard to understand this paper. Captions of Figures 1,2, and 3 are very vague. Authors need to provide more detail. This paper needs more iteration. In addition, there are issues with experimental designs and also the results.

Relation to Prior Work: Yes the differnece is clearly mentioned.

Reproducibility: Yes

Additional Feedback:


Review 2

Summary and Contributions: This paper proposed a geometric method for Boolean Tensor Decomposition. The method sequentially identifies the rank-1 basis components for a tensor from a geometric perspective. Experiments showed its effectiveness in reconstruction accuracy and efficiency on running time. The performance of the proposed method is impressive based on the claims in the paper.

Strengths: The paper tries to provide theoretical proofs and descriptions to endorse the correctness of the proposed method, and provide experimental support using both synthetic data and real world data. The impact of the work seems to be significant compared with existing Boolean Tensor Decomposition methods, as the proposed method has a better accuracy and less cost in running time. Although this method is mostly algorithmic, I believe it is relevant to the NeurIPS community as it provides a state-of-the-art way to solving an important problem.

Weaknesses: To the best of my understanding, the proposed method is actually a greedy algorithm. It would be good if the paper provides theoretical analysis on how close the greedy result is to the ground truth. But I guess it does not affect the significance of the paper from the perspective of experimental results.

Correctness: No obvious error was found.

Clarity: The paper is mostly well written. Detailed comments: 1. Figure 1 is not clearly described. It is also not self-explainable. 2. Typo: line 293, “coutns” -> “counts”

Relation to Prior Work: The paper provides clear discussion for related works. However, I have one doubt regarding the claim at the end of Section 2: “none of the existing algorithms are designed to handle the HBTD problem for higher order tensors”. From what I understand, the LOM paper actually describes the method for higher order tensors, while their implementation was only for 3-order tensors.

Reproducibility: Yes

Additional Feedback: The first step of GETF is to reorder the indices of the tensor into a (k-1)-LTL tensor by IRT. Although the paper provides the computational complexity analysis, I actually have doubts on the efficiency of this step. Does it need all permutations?


Review 3

Summary and Contributions: I have read the rebuttal and my concerns have been partially addressed. || Tensor Decomposition (BTD) factorizes a binary tensor into the Boolean sum of multiple rank-1 tensors like CP decomposition. In this work, the authors proposes an algorithm that sequentially identify rank-1 component via geometry (Geometric Expansion for all-order Tensor Factorization). A theoretical analysis on the validity of the algorithm is provided as well as experiments on both synthetic and real-world data structures, which shows the efficiency of the method.

Strengths: The works apply the recent geometric advances in matrix factorization to tensor decomposition. It is notable that it provide an algorithm that it is not limited to 3d order tensors.

Weaknesses: Geometry-based boolean matrix factorization has been proposed recently and is based on finding the closest Upper Triangular Like matrix possibly with direct consecutive 1’s. Then based on the geometry of such matrix, the pattern of the decomposition can be retrieved from the medium column or row with successive expansion. Once the first component is found, the same methodology is applied to the residual. Even in matricial case, finding a good permutation toward the Triangular shape rely yet on heuristic. So one of the challenges is finding the right permutations up to a (k-1) LTL tensor and then its closest flat 2-LTL tensor, for which the geometry is analogous to that of Upper Triangular matrix. As the proposed algorithm to solve efficiently the determination of a flat 2-LTL tensor is based on the closest plane, it rely on having effectively a (k-1) LTL tensor. The existence of such tensor for any tensor is not assured, or it may be computationally demanding in very high order to do. A study of the performance with a degraded IRT would be a plus. Moreover, there no comparison with other methods. While alternating optimization methods have not been written as yet for higher-order BTF, they can easily be adapted, as the matricization formula used holds for higher-order tensors (and have been used for real-valued tensor factorization).

Correctness: the claims appears to be correct

Clarity: some examples p-LTL tensors just after the definitions (especially (k-1) LTL) and a better insight on flat 2 LTL tensors would be a plus for the reader.

Relation to Prior Work: yes

Reproducibility: Yes

Additional Feedback:


Review 4

Summary and Contributions: The paper addresses tensor decomposition problem for a special tensor, Boolean tensor. The objective is to factorize a binary tensor into the Boolean sum of multiple rank-1 tensors. The paper proposes an efficient BTD algorithm, namely Geometric Expansion for all-order Tensor Factorization (GETF), that sequentially identifies the rank-1 basis components for a tensor from a geometric perspective. Both theoretical analysis and computational experiments are given to validate the performance of the GETF algorithm.

Strengths: The paper proposes a geometric perspective for identifying the largest rank-1 pattern and seeding the most likely pattern fibers. Mathematical formulation is presented and some theoretical analysis on the validity as well as algorithemic efficiency of GETF in decomposing all-order tensor is also provided. Experimental results on both synthetic and real-world data are given to show good performance in reconstruction accuracy, extraction of latent structures.

Weaknesses: The paper provided experimental results on one synthetic dataset and two real world datasets to validate the efficiency and robustness of GETF. But No comparisons with other methods were given. The characters in figure 7 are too small and almost invisible.

Correctness: The mathematical analysis is sound.

Clarity: Well written

Relation to Prior Work: Yes

Reproducibility: Yes

Additional Feedback:

[Author Response · NeurIPS 2020]

We thank reviewers for their efforts and insightful suggestions. In this work, we proposed an all-way Boolean Tensor
Decomposition (BTD) method, GETF, by incorporating the geometric property of Boolean tensor data. We are
encouraged that reviewers find our theoretical driven method effective and efficient, which provides a state-of-the-art
way to solve an important problem. In the following, we answer the main questions and comments from the reviewers.
We will also polish the paper writing to address all the minor issues pointed out by reviewers.

To **Reviewer #1, Q1** *[...why binary representation is better than count representation...].* We think that both binary
representation and count representation have their own merits and how to select the representation should depend
on applications. In this work, we tackle the tensor decomposition problem in Boolean tensors and thus casting the
crime dataset as a binary representation to indicate whether a crime has happened (1) or not (0) is the best choice to
demonstrate our effectiveness. The proposed methods can also be applied to more general scenarios, such as knowledge
graph, contextual recommender systems, search engine and high-dimensional spatial data modeling. **Q3** *[...What is the*
*convergence tolerance?...].* Task completion for experiments is defined as achieving the convergence criteria, presented
in Appendix line 171. And yes, the number of patterns is fixed as 5. We will polish the writing in the revised version to
make it clear. **Q4** *[...the y-axis in Figures Figure 7C and 7D is the same...]* The y-axis of Fig 7C-E are not the same.
Definition of crime index/dates/counts are in the Appendix 4.2. GETF distinguishes the safe and dangerous regions by
reconstructing the relational patterns of crimes, which forms a Boolean relational tensor data (region-date-year), on
which GETF showed better performance over baselines.

To **Reviewer #1, Q2** *[...Why not showing the scalability of GETF and other baselines...]* and **Reviewer #4, Q1** *[...No*
*comparisons with other methods were given...].* In Fig 6 and Appendix Fig 4, we compared GETF with SOTA baselines
on reconstruction error along with the increase of dimensions on different tensor scale, density (corresponds to non-zeros,
Fig 6A-B,G-H) and noise. Due to the page limit, we have to put detailed scalability experiments on real world datasets
in Appendix 4.1, 4.2 and 4.3 (Appendix Figure 7,10), and we apologize for leading to this misunderstanding.

To **Reviewer #2, Q1** *[...It would be good if the paper provides theoretical analysis...].* Thanks for the suggestion, we
will work on this direction in the future.

To **Reviewer #2, Q2** *[...none of the existing algorithms are designed to handle the HBTD problem for higher order*
*tensors...]* and **Reviewer #3, Q4** *[...While alternating optimization methods have not been written...]* Thanks for
pointing this out, and we agree that LOM can be theoretically extended to higher order. However, LOM's Bayesian
fitting scheme is too heavy on computational cost for most higher-order tensors. Empirically, LOM took an hour to
converge on 3D data with 500 features on each dimension (Appendix Fig 5D). Increasing data dimension usually results
in 2 magnitude of increase in data size, yet LOM will fail to converge. In terms of the alternative optimization, besides
the high space/computation cost by Khatri-Rao product for the matricization of higher order tensor, the updating process
is already above O(n) complexity on each dimensionality. For the noisy tensor, this usually results in excessive updating
for this NP-hard problem. Partition the matricization could save some computations, but with a price of increased space
complexity (Park et al ICDE2017). For fair comparisons, we compared our methods on 2D with MP and 3D with LOM,
representing SOTA methods. But for higher order (4D, 5D), we showed the performance of GETF without comparison
as all baseline methods failed to converge on such moderate sized higher order tensors.

To **Reviewer #2, Q4** *[...Does it need all permutations...]* and **Reviewer #3, Q1** *[...finding the right permutations up to*
*a (k-1) LTL tensor...]* The existence of (k-1)-$LTL$ IRT for any tensor is given in line 154-156, and its uniqueness is
supported by Lemma 3 (proof is in Appendix). Not all permutations of IRT are needed to achieve the (k-1)-$LTL$ tensor.

To **Reviewer #3, Q2** *[...a (k-1) LTL tensor and then its closest flat 2-LTL tensor...The existence of such tensor for any*
*tensor is not assured].* GETF is designed based on the geometric property of Boolean tensor, where in a flat 2-$LTL$
tensor, the largest pattern tensor resides on the $1/k$ segmentation point (Lemma1 Fig 2). Lemma 3 and 4 indicate the
(k-1)-$LTL$ IRT is the closest form to 2-$LTL$ IRT. Even the 2-$LTL$ IRT does not always exist, Lemma 2 indicates when
a tensor is sparse and its largest pattern tensor is distinct, the pattern tensor can be sub-optimally detected by the flat
2-$LTL$ tensor with largest solid overlap with the unique (k-1)-$LTL$ IRT. Noted, there are at most n flat 2-LTL tensors
needed to be considered, here n is the tensor size.

To **Reviewer #2, Q3** *[I actually have doubts on the efficiency of this step]* and **Reviewer #3, Q3** *[...it may be*
*computationally demanding...].* For a $n = m^k$ size tensor, since (k-1)-$LTL$ IRT is unique and there are at most n flat
2-$LTL$ tensors to be considered, the *2_LTL_project* is $O(n)$. The complexity cost of the Pattern_fiber_finding algorithm
is $\frac{m^{k+1}-m}{m-1} + kmlog(m)$. On top of the identified pattern fiber, the *Geometric_folding* (Fig 5, main 3.4, Appendix
3.6) algorithm recovered the rank 1 tensor by applying *Pattern_fiber_finding* sequentially that has a complexity cost
at $\frac{m^{k+2}-m^2}{(m-1)^2} - \frac{km}{m-1} + \frac{k(k+1)}{m-1} + \frac{k(k+1)}{2}mlog(m) \sim O(m^k)$, which explained the computational efficiency of GETF.
Moreover, the additional space complexity for GETF is $\frac{m^k-m}{m-1}$, also $O(n)$. We highlighted the complexity analysis in
main text Section 3.5 and will provide detailed derivation of complexity in the revised Appendix.

[Meta-Review · NeurIPS 2020]

This paper presents a greedy sequential algorithm for decomposing a Boolean Nth-order tensor into a Boolean sum of rank 1 components, using Left-Triangular-like and geometric considerations. The paper includes detailed theory, algorithm development and some experiments. This makes the exposition rather dense but the authors have clearly invested a lot time and effort in the work. The task is interesting and the rank-1 pattern revealing algorithm looks nice and intriguing. The work received divergent scores with the main points of disagreement being the difficulty of following the exposition, the practical need for special methods dedicated to Boolean tensors, and the lack of comparative experiments with other algorithms and with (NP hard) exact minimal decompositions. However, in our opinion the method is interesting enough and potentially-useful enough, and the experiments are illustrative enough, for acceptance. The final paper should do its best to address the clarity / reader accessibility issues and to reinforce the experimental comparisons.